# Soil moisture products consistency for operational drought monitoring in Europe

Jaime Gaona[1], Davide Bavera[2], Guido Fioravanti[3], Sebastian Hahn[4], Pietro Stradiotti[4], Paolo Filippucci[1], Stefania Camici[1], Luca Ciabatta[1], Hamidreza Mosaffa[5], Silvia Puca[6], Nicoletta Roberto[6], Luca Brocca[1]

[1]Research Institute for Geo-Hydrological Protection, National Research Council, Perugia, 06126, Italy.
[2] Arcadia SIT, Milano, 27029, Italy.
[3] Joint Research Center, Ispra, 21027, Italy.
[4] Technische Universität Wien, Department of Geodesy and Geoinformation, Vienna, 1040, Austria.
[5] University of Reading, Department of Geography and Environmental Science, Reading, RG6 6UR, United Kingdom.
[6] Italian Civil Protection, Rome, 00193, Italy.

*Correspondence to*: Jaime Gaona (jaimegaonagarcia@cnr.it)

**Abstract**

The roadmap to enable operational soil moisture (SM) monitoring for meteorologic and hydrological early warning depends on the capabilities of the available remote sensing and modelling products. Since each type of soil moisture product shows specific strengths and limitations due to their technical restrictions over certain environments, the detection of impactful anomalies across a wide range of conditions and scales is often challenging and incomplete without a combination of complemental data types of sufficient resolution, revisit time and coverage. This study evaluates the capabilities of SM products of different nature and their compatibility for combination, with special attention to their uncertainties in spatial consistency and in residual trends. While the first has been often revisited to validate remote sensing and modelling products against in-situ data, the last is often overlooked in studies addressing SM changes despite its potential to disrupt the outcomes.

To meet the demands of operational monitoring this study evaluated three SM products: (1) the Satellite Application Facility on Support to Operational Hydrology and Water Management (H SAF) active Advanced SCATterometer (ASCAT)-derived dataset, (2) the passive subset of the European Space Agency (ESA) - Climate Change Initiative (CCIp), and (3) the modelled dataset from the European Drought Observatory (EDO). The analysis was carried out over Europe in the period 2007-2022 at 10-day temporal scales and 5 x 5 km spatial sampling. First, Pearson's correlation coefficient (R) is used to measure the correspondence between H120, H121, CCIp, and EDO SM products. Then triplets of the active, passive and model-based products are applied triple collocation analysis (TCA) to assess their performance based on TCA metrics such as the correlation, error variance, sensitivity and signal to noise ratio.

We obtained that these popular well-validated datasets are increasingly capable in view of the notable TCA scores obtained but still subject to patches of spatial inconsistency and residual trends when compared against in-situ SM data of the International Soil Moisture Network (ISMN). These uncertainties have minimal impact on drought monitoring in most of Europe, except in snow-prone regions and for the assessment of long-term soil moisture trends used to design climate

adaptation policies. Furthermore, each type of soil moisture product prevails in terms of triple collocation scores over the others under specific environmental conditions of the European continent. In view of the synergies shown by the active and passive remote sensing and the modelled SM estimates, two merged products are proposed and tested against the in-situ data.

The merging of the products is conducted by combining the various products based on weights calculated proportionally to the R_TCA scores of the triplets equalized in dynamic range matching their cumulative distribution functions. Results indicate that combining H SAF ASCAT, CCIp and EDO equals or surpasses the spatial and temporal consistency of the individual SM products alone, even when only the near-real-time products of H SAF ASCAT and EDO are combined. The evaluation of the trends of the individual products also indicates that small residual trends remain despite the improved filtering of the

uncertainties, but given their differing sign of the trend, once combined into merged products can provide improved temporal stability of the series. Thus, merging remote sensing and modelled SM products enhances spatial consistency, resolution, temporal coverage, and near-real-time capabilities for better European-scale drought monitoring, strengthening the early warning and risk management systems devoted to improving societal and environmental resilience.

**Keywords**

 Soil moisture, Remote sensing, Model Spatial validity, Residual trends, Drought,

## 1    Introduction

Soil moisture (SM) is a key state variable of the water cycle, fundamental in the study of climate change impacts. SM anomalies are early indicators of altered conditions in both the hydrological domain (Ford et al., 2015; Brocca et al., 2016; Li et al., 2023)

and many critical zone processes (Seneviratne et al., 2010; Green et al., 2019; Bolten and Crow, 2012). Therefore, characterizing SM dynamics over time and space was encouraged long ago (Owe et al., 1999) to better understand the implications of SM changes and their pace of alteration on related processes. Traditionally such analyses were conducted in field studies. However, alternatives are necessary to ease the systematic monitoring of SM hindered by the heterogeneous nature of the variable (Wilson et al., 2004; Zucco et al., 2014) and the persisting lack of funding for SM observation networks

(Dorigo et al., 2021). The emergence of remote sensing (RS) sensors and missions (Schmugge, 1983; Wigneron et al., 2000, Entekhabi et al., 2010) and the rapid development of SM-capable modelling tools (e.g. Sheffield and Wood, 2007; Dirmeyer et al., 2006; De Roo et al., 2000) enabled widespread use of SM data for earth systems analysis (Ochsner et al., 2013).

Two approaches have primarily dominated SM RS technologies: active and passive microwave instruments. Active sensors

detect the reflection of the emitted electromagnetic radiation (radar) while passive ones detect naturally emitted microwave radiation (radiometer) (Schmugge, 1983). Early satellite missions carrying active sensors were not primarily designed for soil moisture detection but proved useful for this purpose beyond their initial meteorological scope (Loew et al., 2013). Such pioneering satellite missions with active sensors include the series of European Remote Sensing Satellites (ERS-1/-2) and the series of Meteorological Operational Platforms (Metop-A/-B/-C). The Satellite Application Facility on Support to Operational

Hydrology and Water Management (H SAF) provides surface SM (SSM) estimates derived from ASCAT on-board the series of Metop satellites since 2008 with near-real-time operability (Albergel et al., 2012).

The passive sensors include the Advanced Microwave Scanning Radiometer-Earth Observing System (AMSR) mission onboard the Aqua satellite launched in 2002 (Njoku et al., 2003) and the SM-dedicated European Space Agency (ESA) Soil
Moisture and Ocean Salinity (SMOS) mission of 2009 (Wigneron et al., 2000). Their decisive contributions stimulated the systematic use of L-band passive data, which are less prone to interferences among different microwave bands (Kim et al., 2013). With the experience of dedicated active and passive missions, the Soil Moisture Active Passive (SMAP) mission initiative of NASA (Entekhabi et al., 2010) aimed to take advantage of combining active and passive sensors. Unfortunately, the failure of the SMAP radar soon after launch left this ambition to merging initiatives like the ESA Climate Change Initiative
(CCI) Soil Moisture Version 08.1 (Gruber et al., 2019), whose passive subset is used here (i.e. CCIp).

The distinct capabilities of the active and passive RS SM datasets soon became object of interest (Scipal et al., 2008). To support the SM retrieval difficulties of one technique with the advantages of the other, multiple studies explored the combination of either data of different RS types or of RS with modelled data, often applying first the scaling of the cumulative
distribution function (Reichle and Koster, 2004). Since then, different merging methods have been proposed, from simple equal weighting to least-square framework that assign weights based on error variances (Yilmaz et al., 2012). These developments have led to the release of combined passive and active global SM datasets (Liu et al., 2012), such as CCI. Combined products reportedly outperform single-source products for SM evaluation (Dorigo et al., 2015; Wang et al., 2021) while still inheriting some limitations of active and passive retrievals under challenging environmental conditions.
In parallel to the development of RS SM datasets, notable progress has been made in modelled SM products, which are increasingly used and evaluated (Beck et al., 2021). From land surface models to conceptual models like LISFLOOD (LF) (De Roo et al., 2000), different modelling schemes have been widely incorporated to meteorological forecasting, reanalysis and monitoring protocols (Van der Knijff et al., 2010). In particular, LF was adopted as the primary tool for providing near real-
time flood risk assessment at continental scale for European Flood Awareness System (EFAS) and drought monitoring in the European Drought Observatory (EDO) (Cammallieri et al., 2015). The flexibility of models is beneficial to evaluate SM sensitivity to the many factors of the complex soil system even under scenarios (Vereecken et al., 2016). However, their high demand on data, intricate parametrization and underlying assumptions may degrade the reliability of their estimates (Fatichi et al., 2016; Samaniego et al., 2010). Conversely, models can generate SM estimates even under conditions challenging for
RS technologies but struggle to characterize relevant features of SM processes such as the heterogeneity.

Despite the increasing capabilities of RS and model-based datasets, the meteorological and hydrological early warning systems demand products ensuring reliable monitoring across a wider range of scales and environmental conditions than the ones the current products alone can provide. For instance, active remote sensing products like ASCAT are known for their high temporal

resolution but may struggle in densely vegetated areas (Wagner et al., 2013). Conversely, passive remote sensing products such as the SMOS and SMAP can be affected by radio frequency interference and surface roughness (Entekhabi et al., 2010). Meanwhile, model-based products reliability can be highly impacted by the quality of the input data, which varies across regions (Samaniego et al., 2013). Therefore, since the characterization of impactful anomalies across some of the environments and scales (e.g. headwaters, densely forested areas, semi-arid areas) that are more relevant for the monitoring is challenging or incomplete without combining data, merging complemental products is beneficial and due to it favoured by both monitoring agencies (e.g. EDO) and the scientific community (Peng et al., 2017; Liu et al., 2011). For instance, the combination of active and passive remote sensing data has been shown to improve SM estimates by leveraging the strengths of both data types (Dorigo et al., 2015; Liu et al., 2012). Therefore, this study pays special attention to the compatibility of different SM products and their combined potential, with a focus on their implicit uncertainties.

Every SM product used for monitoring, whether from RS, models or merged origin undergo validation using a variety of protocols, including the check for validity against the nearby in-situ data (Al-Yari et al., 2019) of the International Soil Moisture Network (ISMN) (Dorigo et al., 2011; Dorigo et al., 2021). However, validation often reveals limitations in coverage, continuity and scale (Loew et al, 2013; Peng et al., 2017). A critical aspect of validation that requires major attention is identifying spurious tendencies (Gruber et al., 2020; Wagner et al, 2022) that can affect the interpretation of the spatiotemporal features of SM data in the context of climate change. Beyond prominent works that proved fundamental reporting change in SM series (e.g. Dorigo et al., 2012; Albergel et al., 2013a; Feng and Zhang, 2015; Cammalleri and Vogt, 2016), multiple studies, even when proposing effective approaches to identify trends, may have reported tendencies that may be artifacts of the SM series. Despite the critical role that uncertainties in SM datasets can play in altering the outcomes of a study (e.g. in trend analysis), explicit analyses of the uncertainties of the data inputs (e.g. residual trends) and their propagation into results have always been rare. Only a few studies have incorporated this crucial step despite its relevance, whether from the early days of pioneering SM products (e.g., Künzer et al., 2008; Draper et al., 2009; Liu et al., 2012) or in recent times with new product versions, improved processing methods or when ingesting new data (e.g. Karthikeyan et al., 2017; Zwieback et al., 2017). Key studies on this topic reported temporal instabilities in SM series (e.g. artificial trends Dorigo et al., 2010; temporal instability Albergel et al., 2013b; or inherited parametrization uncertainties, Crosson et al., 2005). Therefore, specific audit of the uncertainties of the data is required (Peng et al., 2021a; Brocca et al., 2017b).

Consequently, at least these matters require thorough revision before proposing a systematic application of SM products for the exhaustive detection of SM evolution demanded by operational monitoring. This study evaluates three types of SM data: passive RS, active RS and model-based of near-real time capabilities, to better explore their individual and combined consistency and suitability for the operational monitoring of SM. The following activities are targeted:

- Assessing the correlation of ESA CCIp, H SAF remote sensing ASCAT-SSM-CDR-12.5km, including both Version 7 (H120) and Version 8 (H121), with the model-based EDO SM data.

- Discussing the suitability of product merging of active and passive RS SM with model-based SM for operational monitoring across a range of environmental conditions wider than that of the one of each product alone, focusing on near-real time product capabilities.

- Evaluating the performance of the active and passive RS SM, model-based SM and merged SM products against in-situ observation of the ISMN in Europe.

- Describing the trends of these diverse SM datasets and their combination and discussing their specific performance and its impact on trend detection, which has implications on the definition of drought severity and duration.

## 2    Study area

The study focuses on Europe with additional coverage of the areas surrounding the Mediterranean basin. EDO includes data produced by a defined setup of the hydrological model LF whose domain covers almost the whole European continent and the Mediterranean region, approximately in between latitudes 25 to 72.5°N and longitudes 25°W to 50°E. Most RS products are of global scope, and therefore suitable for multiple-scale analysis from the global scale to the scale of their spatial resolution. The geography of Europe comprises a wide range of features across the range of scales covered by the spatial resolution of the RS SM products. However, some RS SM products are somehow limited at high latitudes (latitudes higher than 60ºN) due to physical characteristics of the environment (permanent frozen soil and snow cover, water bodies and vegetation interference) results of the boreal belt (60°-72.5°) are included in the study to illustrate the challenges of SM monitoring over such areas. The area of study is displayed using Lambert Azimuthal Equal Area (EPSG: 3035) centred at 50°N, 15°E. (Fig. 1).

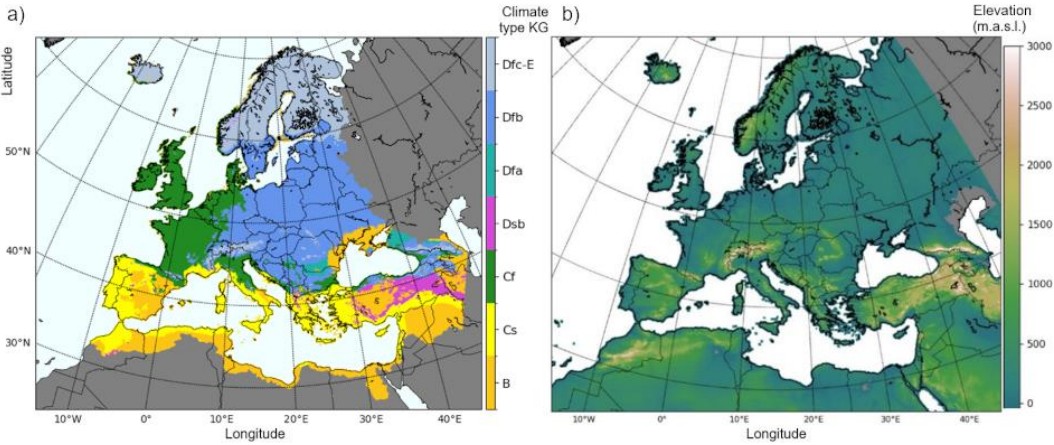

**Figure 1: a) Climatic classification based on Köppen-Geiger climate types of Europe and the circum-Mediterranean region (adapted from Beck et al., 2018; CC BY 4.0). Not coloured areas represent climate types excluded from analysis. b) Elevation map based on ETOPO_2022 by NGDC NOAA (MacFerrin et al., 2024; CC0 1.0).**

The scope at the European scale grants the existence of multiple climatic regions of particular environmental characteristics (Fig. 1b) within the study area. Not only in terms of latitude, from the tundra of northern Scandinavia to the arid and semiarid

regions of the Mediterranean basin, but also from sea level to alpine altitudes, Europe offers a wide range of climates (Fig. 1a: Map of climatic areas based on the classification of climates of Köppen-Geiger (Beck et al., 2018). At least three out of five of the main climatic domains of this classification can be found across the continent, the type *B* climate of semi-arid to arid regions defined by precipitation, and two temperate climates determined by the annual temperature range: the type *C* of low annual temperature range modulated by sea influence and the type *D* of wide annual temperature range prevalent in the continental inlands. Other geographical aspects beyond climatic zoning such as land cover have not been considered but play a role in uncertainties of the RS SM data retrieval such as biomass content or dense vegetation (Pfeil et al., 2018; Ma et al., 2019; Ikonen et al., 2018) and are only secondary object of comments in discussion. (Fig. 2: Map of ISMN networks with land cover).

## 3    Materials and methods
### 3.1    Remote sensing soil moisture data

#### 3.1.1    Active microwave soil moisture

*H SAF soil moisture products*
The C-Band real aperture radar system of ASCAT onboard Metop-A satellites since 2006, Metop-B since 2012 and Metop-C since 2018, collect active microwave data at sun-synchronous near-polar orbits that is processed using the TU Wien SM retrieval algorithm to generate the H SAF SSM products. The products used here ASCAT-SSM-ICDR–12.5km-v7 (H120) (https://hsaf.meteoam.it/Products/Detail?prod=H120) and the upcoming ASCAT-SSM-CDR–12.5km-v8 (H121) cover the study period 2007-2022 and incorporate the last improvements on signal processing and correction (Hahn et al., 2017). This remarkable length, continuity and coverage of the ASCAT-derived H SAF SSM products have popularized RS SSM for multiple applications (Brocca et al., 2017). The H SAF products H120 and H121 used here have a spatial sampling of 12.5 km arranged on a Fibonacci spiral grid at a spatial resolution of 25 x 25 km. SSM is expressed as degree of saturation (0% dry soil, 100% fully saturated soil) of the first few centimetres of the soil (< 5 cm) as water volume present in the soil relative to pore volume (Wagner et al., 1999; Naeimi et al., 2009).  The combination of the data from the different satellites MetOp A, B, C (currently only B and C) covers the entire globe every few days.

#### 3.1.2    Passive microwave SM based on C-band and L-Band retrievals:

*ESA CCI passive soil moisture dataset*
ESA CCI passive dataset (CCIp) is the subset of ESA CCI SM v08.1 (Dorigo et al., 2017; Gruber et al., 2019) (https://climate.esa.int/en/projects/soil-moisture/) based on merging of passive sensors only. The data is provided globally at a sampling of 0.25° x 0.25° with more frequent spatial gaps in the early years of the dataset (Loew et al., 2013). Alpine or boreal regions and densely forested areas show spatial and temporal gaps of the retrievals due to frozen soils or the canopy cover attenuation (Dorigo et al., 2017). The data at daily temporal resolution is available from November 1978 to the end of 2023. The merging is conducted on the basis of the signal-to-noise ratio and scaled against SM dynamic ranges of GLDAS-Noah v2.1 land surface model (Rodell et al., 2004) and break-adjusted (Preimersberger, 2020).

### 3.2 Model-based soil moisture data

#### 3.2.1 The European Drought Observatory (EDO)

EDO (https://edo.jrc.ec.europa.eu) provides LISFLOOD (LF) model-based SM estimates. LF is the distributed rainfall-runoff model initially developed for flood forecasting by the Land Management and Natural Hazards Unit of the Joint Research Centre (JRC) of the European Commission. The first two layers (corresponding to the root depth) of the three layers provided by the model when simulating the water balance of the catchment are considered for the commutation of the SM index. The dataset has a 5 × 5 km grid cell size. To perform all the analyses, H120 / H121 and CCIp SM products are regridded to this finer spatial resolution of EDO. The dataset spans from 1991 to present at daily temporal resolution but is commonly provided at the 10-day period corresponding to the 1st, 2nd and 3rd third of the month.

### 3.3 In-situ soil moisture data

#### 3.3.1 The International Soil Moisture Network (ISMN)

The International Soil Moisture Network (ISMN, https://ismn.earth/en/, Dorigo et al., 2021) is the collective initiative supported by ESA to compile the data of multiple networks observing SM around the globe originated for various purposes. Since the SM data became essential for RS, the ISMN aims to favour the harmonization of the SM observations. As of May 2024, the ISMN database hosts 80 networks data all over the world, including the 26 networks across Europe included in this study with data available in the study period 2007-2022 (Table S1, supplementary material). The networks comprise a diverse but uneven range of climates and land cover (Table S1, supplementary material). The assorted scales and measuring settings of the included ISMNs are of challenging spatial representativity compared to the distributed RS data (Gruber et al., 2013).

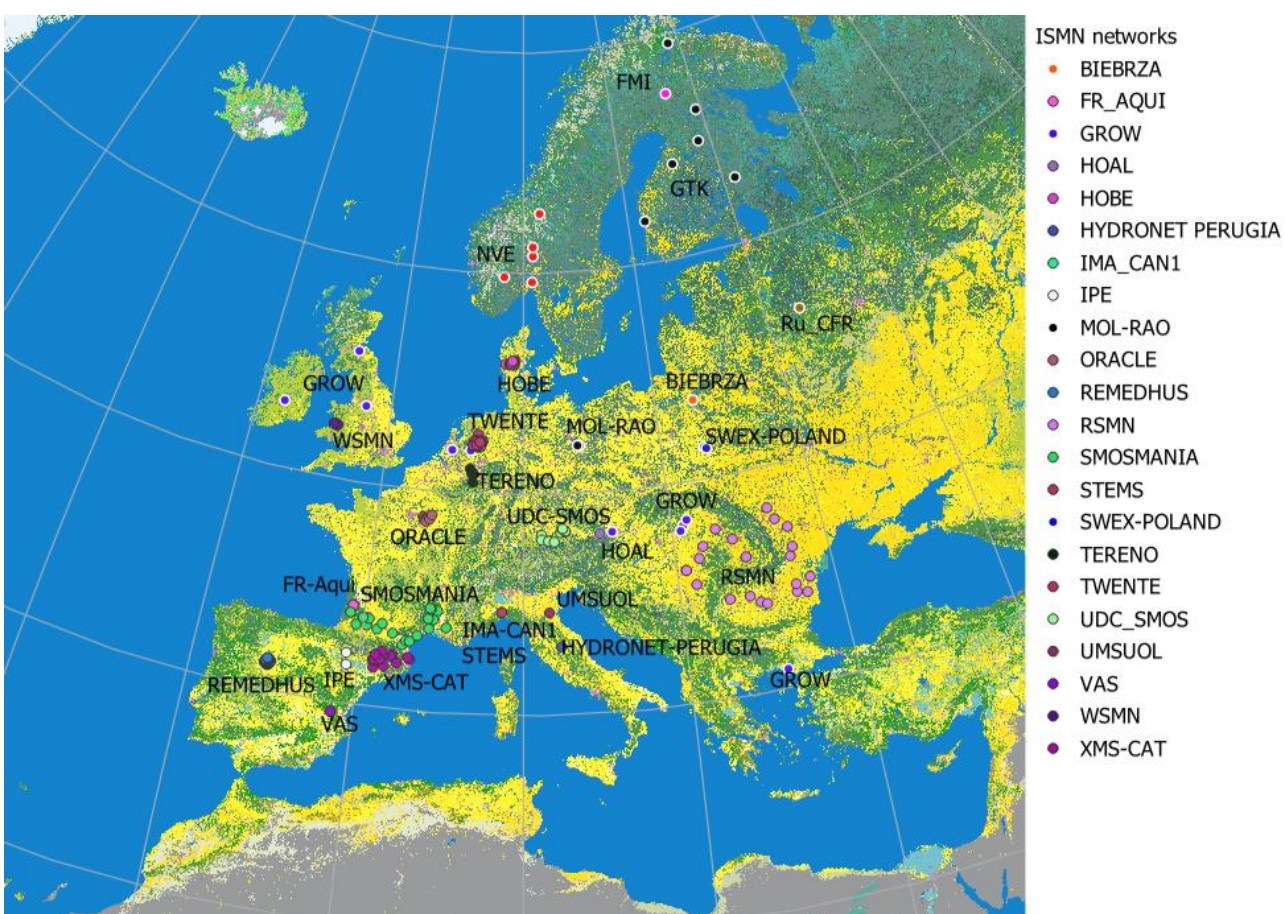

**Figure 2: Location of the SM networks of the ISMN initiative in Europe superimposed over ESA CCI Land Cover 300m 2015 (Kandice et al., 2023). Yellow/green colours comprise the multiple land cover classes without/with significant vegetation effects on the retrieval of soil moisture using remote sensing. Urban areas are coloured in pink.**

### 3.4 Methodology

#### 3.4.1 Preprocessing

Initial datasets feature diverse spatial and temporal scales. For this reason, both spatial and temporal pre-processing of the datasets is required. In spatial terms, the datasets are re-gridded to the reference spatial grid of EDO of 5 x 5 km using the search of nearest neighbours of KD-Tree algorithms. In the case of H120 and H121, a transformation from the original Fibonacci swath geometry used by ASCAT to the regular grid geometry of the other datasets is required. The temporal time

step defined for the analyses is the 10-day time scale, which is the time scale followed by the EDO. Each of the three reference dates per month arranged in this tri-monthly basis represent the average conditions of a third of the month (i.e. from the 1[st] to the 10[th], from the 11[th] to the 20[th,] and from the 21[st] to the 31[st] of a month). All initial values within each of third of the month at daily time step are aggregated to the reference date of the corresponding interval. In the case of H121, the aggregation to

the 10-day time scale is computed by directly aggregating the hourly-scale datasets within the 10-day period during the
computation of the Soil Water Index (SWI) from the original SSM given in degree of saturation. The computation is based
on the exponential smoothing filter (Wagner et al., 1999b; Albergel et al. 2008) that converts the surface soil moisture
saturation degree, ms$(t)$, into the SWI$(t)$ (Eq. (1)):

$$SWI(t) = \frac{\sum_{i}^{n} m_s(t_i) e^{-\left(\frac{t-t_i}{T}\right)}}{\sum_{i}^{n} e^{-\left(\frac{t-t_i}{T}\right)}}$$

(1)

where, regardless of the units of product, the soil moisture retrieval at time $t_i$ is $SM_{sat}$, the time lag introduced with the filter is
$T$, and $t$ represents the 10-day time step. $T$ was set to 10 days for all the products. The SWI ranges between 0 and 1 from dry
to wet conditions. More than 3 retrievals in the 10-day interval $t$ were prescribed for calculating SWI, following Pellarin et al.,
(2006). Due to the exponential filter smoothing effect (and delay) of $ms(t)$ the range of SWI$(t)$ varies in a narrower range than
the [0,1] degree of saturation range.

The 10-day time scale is used for the intercomparison and evaluation of the products against the ISMN while the monthly scale
is adopted for the trend analysis of SM anomalies. The SM anomalies are computed by removing the seasonal cycle, which is
defined by the mean SM value of each month of the year. These twelve mean SM values are obtained by averaging all 10-days
SM within each calendar month occurring along the full length of the study period (2007-2022) without seasonal focus. In the
vast majority of the areas the number of values involved in obtaining the mean SM of the month surpasses tens of values per
month, with only a few alpine or boreal snow-prone areas being calculated with a few values per month. The SM anomaly is
finally calculated as the ratio of deviation of SM of any month from the mean SM of that month.

The different product additionally provides metrics of the error characteristics to identify the areas and periods affected by
relevant impactful factors such as snow cover. The flag scheme of ESA CCIp exemplifies the detailed procedures devoted to
distinguishing when the data is subject to further filtering. Specifically, ESA CCIp assesses snow and frozen soil using both
temperature and freeze-thaw conditions (via Ku-, K-, and Ka-band retrievals, as noted in the CCIp ATDB guide) Consequently,
considering the notable relative importance of the snow cover factor over the others factors in our analysis over Europe, we
have applied this specific and restrictive snow cover flag of ESA CCIp as mask to the data coverage of the other products of
the analysis over the snow-prone regions.

Conversely to the case of the distributed datasets, the point data from the ISMN database has not been severely restricted with
flags due to the general consistency of most stations in common environmental conditions and the scarcity of stations with
severe indications of uncertainty from the flags. In such cases, the inclusion of the stations was decided based on the existence
of alternative stations with similar environmental conditions, and if not available included based on the criteria of consistency
between the factors of uncertainty causing the flag and the characteristics of the environment. Multiple areas where snow and
icing processes are frequent are barely observed in situ, and consequently, even despite the seasonal uncertainties, RS products

provide much more coverage of these areas than the few ISMN stations over these areas. Therefore, including all ISMN networks and data available was the decision adopted to ensure enough data for validation over every climatic type and ensure the representativity of a wide range of observed soil moisture conditions. All stations with available data at the uppermost depth of interest were included. Nonetheless, the data series from each station were filtered to remove unrealistic data indicated by flags for discontinuities, out-of-range values, and other artifacts, regardless of uncertainties from location, instrument type, or other factors.

### 3.4.2 Performance metrics

The Pearson's correlation coefficient (R) is used to quantify the correspondence between H120 and H121, CCIp and EDO SM products and the ISMN SM data. The triple collocation analysis (TCA) (Stoffelen, 1998; Scipal et al., 2008) is also used to estimate the random error variances of the collocated H120 / H121, CCIp, and EDO triplets. The TCA model assumes linearity of SM retrievals, stationarity of signal and independence of errors from signal or in between product errors (Gruber et al., 2016; Massari et al., 2017; Filippucci et al., 2021). However, since the purpose of the study is not to evaluate the sensitivity of the assumptions of the TCA model we focus on the adequacy of the selected SM products for triplets based on their acknowledge independence and consistency. The LISFLOOD model is not used for the processing or validation of neither the passive nor active RS products that complete the product neither on the curation of the ISMN data. The ASCAT active RS SM upgrades on the TU Wien Change detection algorithm under the release of H120 and recently H121 are supposed to resolve the increasing trend and consequently can be considered only marginally non-stationary. CCIp uncertainties are to a great extent already resolved in the dedicated processing methods developed for the error characterization of data from different passive SM missions (i.e. The Land Parameter Retrieval Model, LPRM). The same assumption can be applied to the residual trends of EDO and CCIp SM datasets that are of a magnitude likely more relevant for long-term analysis rather than for its impact on the TCA assumptions. Therefore, given the sufficient data quality and independence of the three selected products, we can assume the classic model of TCA as suitable for our aim and expressed as in Eq. 2 below:

$$X = \alpha_X + \beta_X \theta + \varepsilon_X$$

$$(2)$$

where the spatially and temporally collocated datasets are compiled in the dataset $X \in$ [H120 or H121, CCIp, EDO], the soil moisture is $\theta$, and $\alpha X$ is the systematic additive error behind the offset between the temporal and the true mean of $\theta$. The $\beta X$ is the coefficient of multiplicative biases of $X$, and noise is represented by $\varepsilon X$. Even in the case of SM whose random error can depart from a gaussian distribution, the error variance can be expressed as in McColl et al. (2014) (Eq. (3)):

$$\sigma_\varepsilon = \begin{bmatrix} \sqrt{Q_{11} - Q_{12}Q_{13}/Q_{23}} \\ \sqrt{Q_{22} - Q_{12}Q_{23}/Q_{13}} \\ \sqrt{Q_{33} - Q_{12}Q_{23}/Q_{12}} \end{bmatrix}$$

$$(3)$$

where $Q_{ij}$ is the covariance of dataset $i$ against $j$, which leads to the expression of TCA correlation scores $R\_TCA$ (Eq. (4)), which is a relative measure against the unknown truth:

$$R\_TCA = \begin{bmatrix} \sqrt{Q_{12}Q_{13}/Q_{11}Q_{23}} \\ \sqrt{Q_{12}Q_{23}/Q_{22}Q_{13}} \\ \sqrt{Q_{13}Q_{23}/Q_{33}Q_{12}} \end{bmatrix}$$

(4)

Additionally to the R_TCA score, auxiliary performance metrics such as the error variance, the sensitivity and the signal to noise ratio defined in Gruber et al. (2016) are obtained from the TCA analysis and occasionally referred to further clarify or discuss the results. The performance of each product is computed by aggregating the scores over the regions occupied by each

300 Köppen-Geiger climate type across Europe (Fig. 1a, (Beck et al., 2018). Furthermore, the TCA approach and its performance metrics are also suitable for merging RS SM products (Gruber et al., 2017), among many options, as described below. 🄾🄱🄹

### 3.4.3    Definition of the products merging RS and model-based SM data

Merging is achieved by combining the SM estimates of the intervening products proportionally to weights based on their

different $R\_TCA$ scores of the TCA. The triplets of TCA generating the $R\_TCA$ scores are equalized in dynamic range matching their cumulative distribution functions (CDF) (Brocca et al., 2010b). The expression to merge the SM product is:

$$SM_{merg2} = -\omega_{HSAF} \cdot SM'_{H\,SAF} + \omega_{CCIp} \cdot SM'_{CCIp} \cdot \omega_{EDO} \cdot SM'_{EDO}$$  (5)

where $\omega_i$ is the relative weight of each product's R_TCA scores, obtained from Eq. (6):

$$\begin{cases} \omega_{H\,SAF} = \dfrac{R\_TCA_{H\,SAF}}{R\_TCA_{H\,SAF} + R\_TCA^*_{CCIp} + R\_TCA^*_{EDO}} \\[2mm] \omega_{CCIp} = \dfrac{R\_TCA^*_{CCIp}}{R\_TCA_{H\,SAF} + R\_TCA^*_{CCIp} + R\_TCA^*_{EDO}} \\[2mm] \omega_{EDO} = \dfrac{R\_TCA^*_{EDO}}{R\_TCA_{H\,SAF} + R\_TCA^*_{CCIp} + R\_TCA^*_{EDO}} \end{cases}$$

(6)

Where $R\_TCA_{H\,SAF}$ $R\_TCA^*_{CCIp}$ and $R\_TCA^*_{EDO}$ are the TCA correlation scores of H SAF (H120 or H121), CCIp and EDO. CDF matching (') is applied in reference to the product without the mark (H SAF H120 or H121) to equalize the dynamic ranges. The two products obtained with this procedure of merging H SAF (H120 or H121), CCIp and EDO or only H SAF

(H120 or H121) and EDO are hereafter respectively denominated 'MERG_h121_3', and 'MERG_h121_2'.

### 3.4.4    Evaluation against in situ data

The evaluation of SM products against in-situ data used all available ISMN within Europe for the period 2007-2022 despite the existing several factors of uncertainty regarding the quality of the data (e.g. representativity or SM range). The ISMN stations whose data availability is shorter than that of the study period were paired to the corresponding equal period of the RS

H120/H121, CCIp, the model-based data from EDO or their combination. All products were also aggregated to the reference 10-days scale of EDO. Pearson correlations to evaluate RS and model-based against the ISMN records were computed by

extracting the corresponding time series of RS and modelled datasets at the locations of the stations of the ISMN networks, defined by their latitude and longitude, using KD-Tree algorithms of nearest neighbours.

### 3.4.5 Trend analysis

In this work, the Mann-Kendall (MK) (Mann, 1945; Kendall, 1948) methodology was considered to evaluate SM anomaly trends by the significance of the monotonic upward or downward trends. The lack of trend is indicated with a valid null hypothesis when data is independent and randomly distributed. We consider a significance level of 0.05, corresponding to values of the statistic $Z>1.96$, to reject the null hypothesis and consider that the trend is significant (Rahmani et al., 2015).

## 4 Results and discussion
## 4.1 Characterizing the spatiotemporal concurrence between SM products
### 4.1.1 Linear correlation analysis

The temporal correlation quantifies the correspondence between EDO, CCIp and H120 or H121. The correlation between CCIp and another product gets the highest values either using H120 (R Pearson median/mean=0.59/0.48) or H121 (0.68/0.58) (Fig. 3b, e). H121 version induces a notable improvement of the scores in respect to H120 also for the correlation with EDO (from R=0.50/0.39 using H120 to R=0.59/0.48 using H121) (Fig. 3a, d). The EDO-CCIp correlation remains intermediate compared to the others (R=0.55/0.51) (Fig. 3c). Results indicate differences between the products over some areas of the continent. Products agree with R>0.7 over the British Isles except for Scotland, Benelux, western areas of Germany, France except the Alps, the Atlantic basins of Iberia and some Mediterranean areas (e.g.: Peng et al., 2021b; Parinussa et al., 2014; Brocca et al., 2011). Multiple other regions in the Mediterranean basin display R>0.5 in line with previous reports (e.g.: Juglea et al., 2010; Brocca et al, 2010b; Duygu et al., 2019). Only continental central and NE Europe show less consensus (R< 0.4) between products, particularly when EDO intervenes.

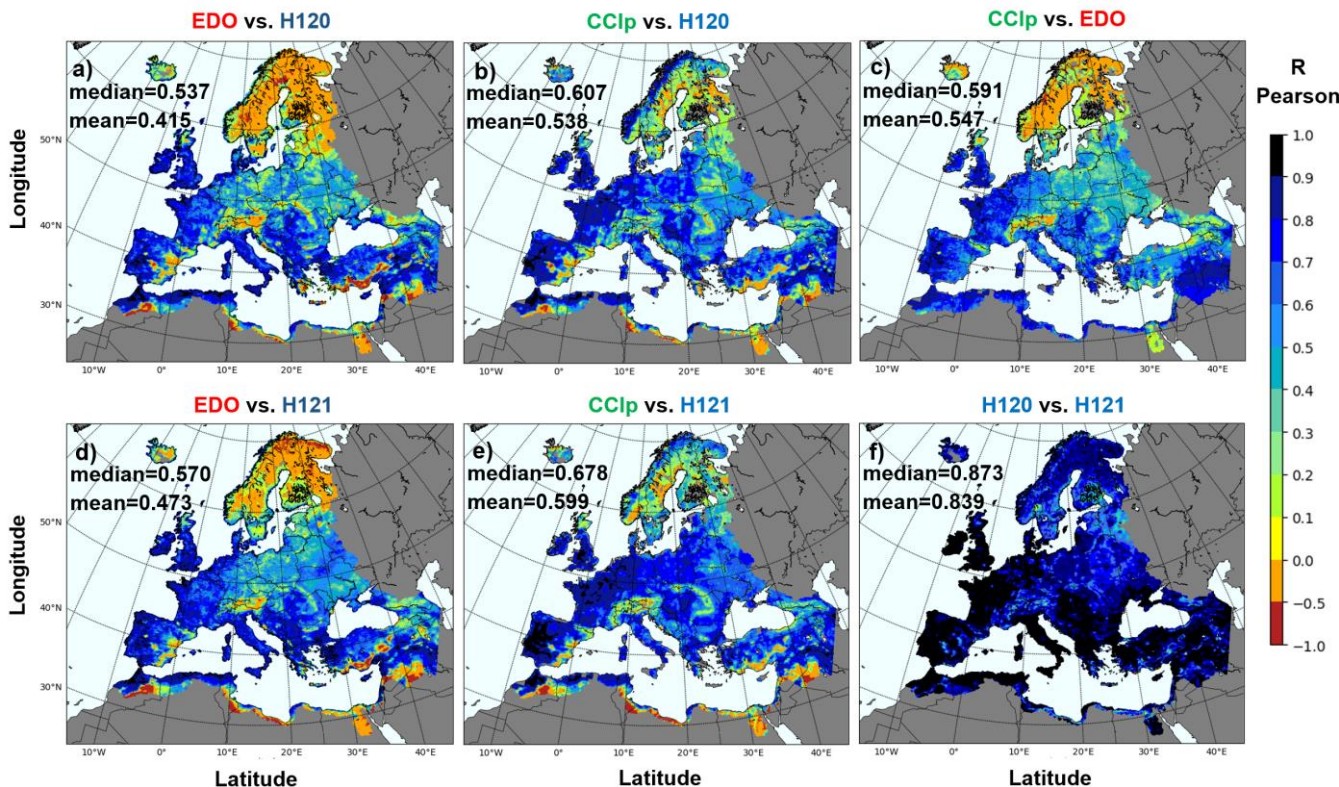

**Figure 3: Map of temporal R-Pearson correlation in between a) EDO and H120, b) CCIp and H120, c) CCIp and EDO, d) EDO and H121, e) CCIp and H121, f) H120 and H121.**

The mismatch between RS and EDO SM data in eastern Europe may be partly attributable to the uncertainty of LF model during winter (Cammalleri et al., 2015), especially since RS SM data is increasingly reliable in boreal areas (Ikonen et al., 2018). The water fraction may also interfere over lake areas of NE Europe (Paulik et al., 2014), but snow cover has been long considered the prevalent cause of blurred determination of SM in *D-E* type climates, including mountains, with higher impact over EDO than over RS data (Laguardia and Niemeyer, 2008).

However, the apparent SW-NE gradient of consistency in between SM products attributed to snow prevalence may be SM regime-related, as it corresponds to the gradient of SM regimes depending on water or energy-dominated conditions (Denissen et al., 2020, Fig. S1). The areas most influenced by westerlies (*C* climate types) of contrasted winter-summer SM regimes are the ones where SM products concur the most. Conversely, *D* climate types of East Europe that tend to sustain water-dominated

SM regimes during summer exhibit the lowest similarity. Such SM-regime implications may require further analysis but have been recognized as impactful at least on the backscattering of active RS SM products (Wagner et al., 2022).

The negative R-Pearson scores shown in Fig. 3 in between EDO and H120/H121, between CCIp and EDO, and to a lesser extent between CCIp and H120/H121 show even negative R-Pearson correlations in snow-dominated areas and thin soils in

mountains and in arid climates. These negative values refer to disagreement between products beyond low performance. The combination express that EDO is the greatest source of such disagreement but differently for northern and southern latitudes: the negative scores induces by the participation of EDO seem to be mostly restricted to snow-dominated regions (compare Fig. 3a and 3b with 3c) while H120/H121 seems to be the primary source of low scores in the arid areas. These results are consistent with the known difficulties of EDO-LISFLOOD in snow-dominated hydrological regimes and of ASCAT H120/H121 over thin soils of arid areas.

Topography (Fig. 1b) is also a principal factor that in combination with arid conditions induces uncertainty in the SM products. Rough areas of Iberia (Escorihuela and Quintana-Seguí, 2016), north Africa, south Greece and Anatolia display low scores likely related to the subsurface scattering over thin soils of either C-band (ASCAT) or L/Ku-band active sensors (McColl et al., 2013; Wagner et al., 2022).

### 4.1.2    Triple collocation analysis

TCA provides an accurate quantification of the correspondence in between SM products and in respect to the unknown reality (Stoffelen et al., 1998; Gruber et al., 2016). TCA results differ from the linear correlation analysis (Fig. 4 vs. Fig. 3). In the triplet of H120, CCIp and EDO, H120 scored in between EDO and CCIp (Fig. $4a_1$-$c_1$). However, H120 seems fairly accurate in line with reports against ERA5Land (Pierdicca et al., 2015a). Therefore, when the improved version H121 is used in the triplet, it leads the scores (Fig. $4a_2$-$c_2$), including over $D$ and $E$ type climates (Fig. 5). CCIp leads the scores in the triplet adopting H120 (Fig. $4c_1$), which was already remarkable because only CCI combined was reported superior to H120 before (Al-Yaari et al., 2019; Fan et al., 2022). CCIp remains second in R_TCA in the triplet including H121, but in this case even EDO may achieve equal or better scores among climate differences. Nonetheless, EDO displays the lowest R_TCA scores in both triplets using H120 and H121, especially in the NE of Europe (Fig. $4a_1$, $a_2$). Multiple studies reported difficulties of models on snowed/frozen areas (Naeimi et al., 2012), but low scores are shown by all products in high latitudes / elevations climate types (*Dfc-E*) (Fig. 5). Our results also indicate poor performance of EDO (range of R_TCA: 0-0.3) over areas of episodic or seasonal snow-cover (*Dfb* and *Dfc-E* respectively from mid-December to mid-March) compared to the moderately good values of the snow-free period (mid-May to mid-October) (R_TCAs: 0.3-0.9).

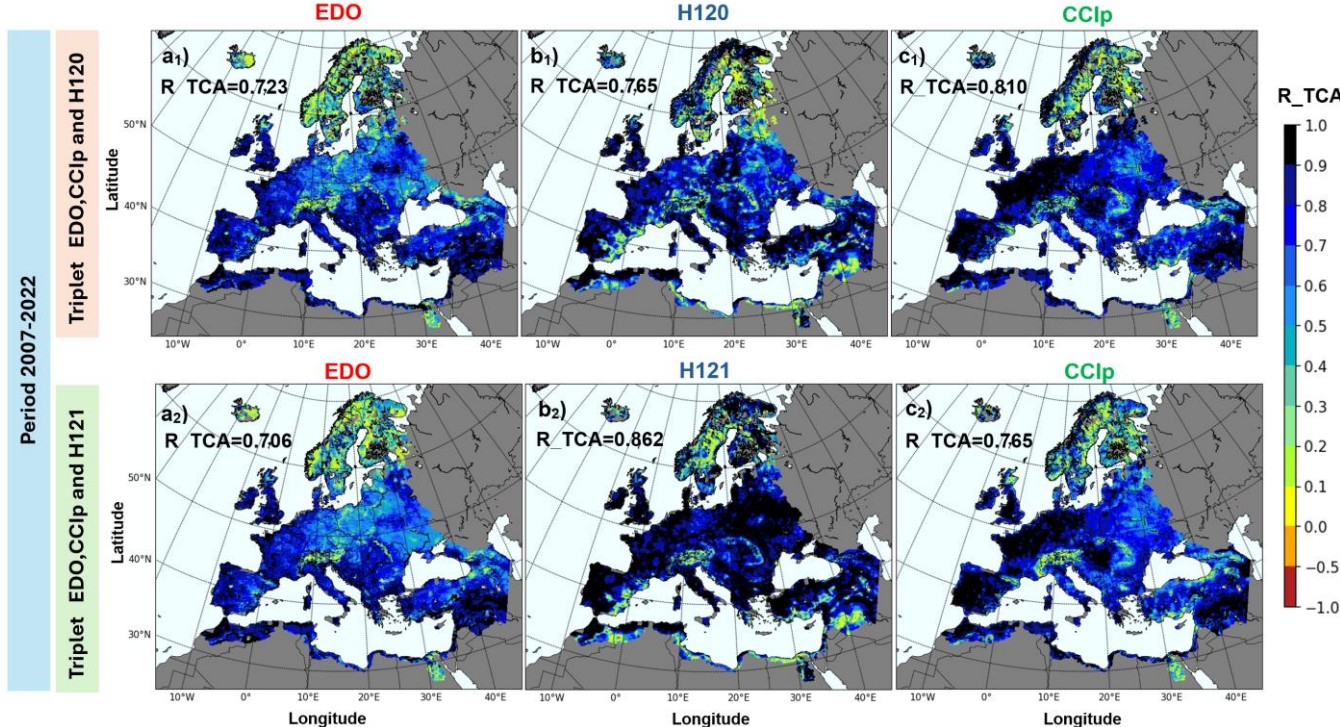

**Figure 4: Maps of R_TCA score of triple collocation analysis of the period 2007-2022 of the triple of the (a₁) modelled-based EDO, (b₁) the active RS H120 and c₁) the passive RS CCIp SM products. The second triplet (a₂, b₂, c₂) just replicates the former with H121 in b₂) replacing the version H120.**

The percentage of areas with values over the sufficient threshold of R_TCA>0.6 differs in between products. H121 champions with more than 65% of Europe over this value and with close to the 40% with R_TCA>0.9, closely followed by CCIp with 60%, while EDO only achieves a 52%. The low values of triple collocation correlations concurring over the snow-prone areas of Scandinavia and snow-caped mountain ranges (e.g. Alps) affect particularly EDO and CCIp, and experienced a significant improvement over snow-prone lowlands of northeastern Europe with the update of ASCAT data from H120 and H121. Beyond

the evidence that EDO model is limited over snow-dominated regimes, CCIp seems to display a bimodal downgrade of scores with low values over Scandinavia and  intermediate ones over the also seasonally snow-dominated but less densely vegetated East Europe suggesting that uncertainties originated from high-latitude and dense canopies may differ from those originated from the snow regime alone (Blyverket et al., 2019).  This result is also supported by the contrast between the low R_TCA of eastern Europe (*Dfb*) (R_TCA range: 0-0.3) compared to the  moderate R_TCA values of some high areas of Scandinavia

where during the snow-free season when temporal snow episodes can also occur like those of *Dfb* in winter. In these cases, Scandinavia R_TCA are much higher than at *Dfb* (R_TCA range: 0.1-0.6), which suggest an additional influence of canopy.

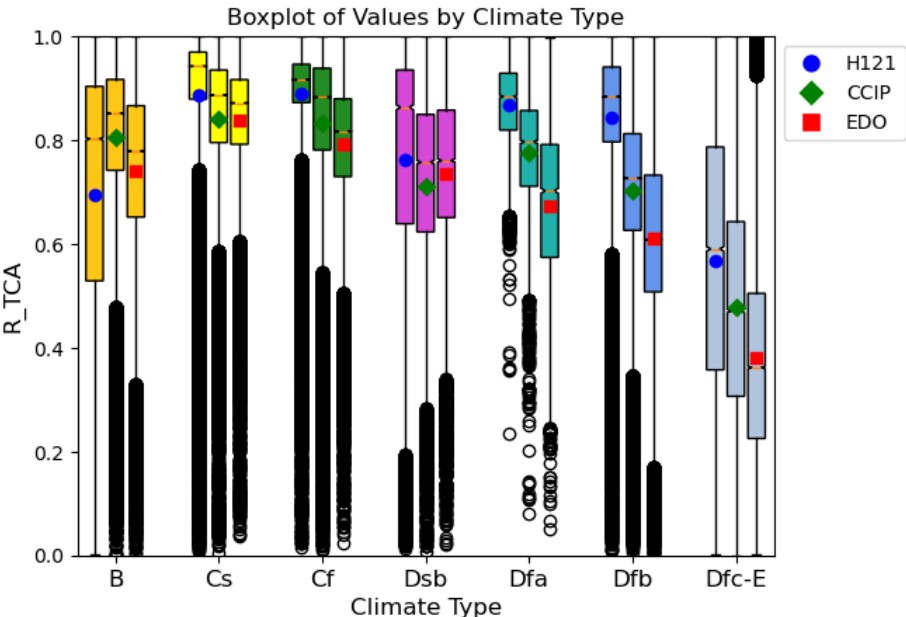

**Figure 5: Boxplots of triple collocation score (R_TCA) of the model-based EDO (boxplots with red square indicating the mean) the active RS H121 (indicated with blue circles) and the passive RS CCIp SM product (depicted in green rhombus) in between diverse types of climates (coloured as Fig.1b).**

The pattern of high consistency over western Europe (type *C* climate) (Fig. 5) has been already reported for EDO, H120 or CCI (LF, Cammalleri et al., 2017; ASCAT, Chen et al., 2018; CCI, ERA-5 Land, Pierdicca et al., 2015b; CCI-ERA-Interim, Deng et al., 2019). Such west-east gradient pattern has been either attributed to coverage according to the number of valid triplets available at a certain location) or subject of discussion about the distinct modes of SM variability in Europe (dominated by seasonal variability in the west but dominated by long-term variability in the east: Fig 11 of Piles et al., 2019). The prominent spots around urban areas due to urban backscattering visible in H120 are solved in H121. This underlines the relevance of updates incorporating processing improvements. However, the reduced signal in SE Spain that corresponds to the reported backscattering of arid areas (Wagner et al., 2022) or alternatively related to the autocorrelation error common in low LAI areas (Dong and Crowd, 2017) remain. Similarly, swampy areas like the Pinsk/Pripet River floodplain in between Belarus and Ukraine that downgrade both H120/H121 and CCIp signals remain better covered by EDO. A bit to the west of this area, a diagonal of values of CCIp R_TCA≤0.8 in between northeast Germany and Ukraine is prone to radio frequency interference (Oliva et al., 2016) as shown by the filtering of SMAP (Mohammed et al., 2016). Hence, none of the products alone can fully characterize SM across Europe with the same accuracy despite their overall good agreement and partial complementarity, even after updates. However, results also indicate the independence in between products which is an important prerequisite to address the generation of combined products, because overlooking interdependencies in the products may also undermine the consistency and reliability demands of operational monitoring.

## 4.2 Suitability of merged products for operational monitoring of SM

The convenience of merging RS and model-based SM datasets to get the full potential of their synergies can be illustrated with the maps of best performing SM product over Europe (Fig. 6). Extensive areas of Europe are dominated by CCIp, mostly in the areas facing west. Remarkably, these areas now better depicted by CCIp were H120-dominated areas in the past (Leroux et al., 2013; Al-Yaari et al., 2014). The prevalence of CCIp is solid in those regions but slightly reduced in area from H120 to the latest version H121 (green areas, Figs 6a, 6b). The eastern part of the continent remains better portrayed by H120 than CCIp, and even better by the H121. EDO that prevailed as best product in north latitudes is primarily replaced there by H121, thus illustrating the notable progress of RS products despite their sensitivity to the challenging freeze-thaw processes (Naeimi et al., 2012) or dense forest cover (Van der Molen et al., 2016; Ikonen et al., 2018). The prevalence of EDO by the coast explained by the limited signal retrieval of RS sensors in the vicinity of the sea (Brocca et al., 2011; Kerr et al., 2012; Portal et al., 2020) becomes also reduced with the new H121 version. Nonetheless, rough, arid or swampy areas of uncertain RS data remain better recognized by EDO, followed by CCIp. In general, most changes when substituting H120 to H121 in the triplets are favourable to H121, whose share of areas as best product increases from similar to CCIp to a dominating 58% (Fig. 6a vs. 6b). CCIp change to H121 contributed more than EDO change to H121 (Fig. 6c) to the dominance of H121 (Fig. 6b). EDO experiments more declined than CCIp in shared area.

The spatial prevalence of CCIp and H120 / H121 over western and eastern Europe respectively agrees with the climatic division in between *C* and *D* climates (Fig. 6b). Intrinsic hydroclimatic differences may be the cause, either as an expression of the distinct SM regimes as identified with self-organizing maps (Markonis et al., 2020; see Fig. 4), as expression of the oceanic vs. continental moisture (Gimeno et al., 2012) or due to distinct SM variabilities (Piles et al., 2019). Ecoregions, which also express hydroclimatic differences, have also evidenced differences in the consistency of SM estimates among products (Mazzariello et al., 2023). The depth of the active and passive RS SM retrievals might be also distinctly sensitive to the dominant rewetting process (Lun et al., 2021; Santos et al., 2022). Even though the reasons behind these differences are beyond our scope, they emphasize the complementarity of the active, passive and modelled SM products. Hence, it is reasonable to develop products combining RS and modelled SM data (Parinussa et al., 2014; Peng et al., 2021b) that favour the best performing product over each area while maintaining a balance according to their performance metrics. The two proposed merged products combine the best of the ASCAT derived products, H121, and EDO ('Merg_h121_2') or H121, EDO and CCIp ('Merg_h121_3'), and their performance is evaluated together with H121, CCIp, EDO against the ISMN data including trends. The reason for not including CCIp in the Merg_h121_2 is to determine if H121 and EDO alone can achieve similar performance to the individual products or the fully combined case, especially considering the increasing accuracy of ASCAT data (from H120 to H121) which is encroaching on areas previously dominated by CCIp (Fig. 6a, vs. 6b). In consequence, the merging aims to explore the compatibility of RS and modelling to fully characterize SM processes even in the areas where RS, models or both struggle. That is why given the effort devoted to developing specific missions for SM observation, the scientific community and agencies emphasize the importance of exploring the data capabilities to their full potential while contributing to advance in both the scientific understanding and the operational monitoring of SM.

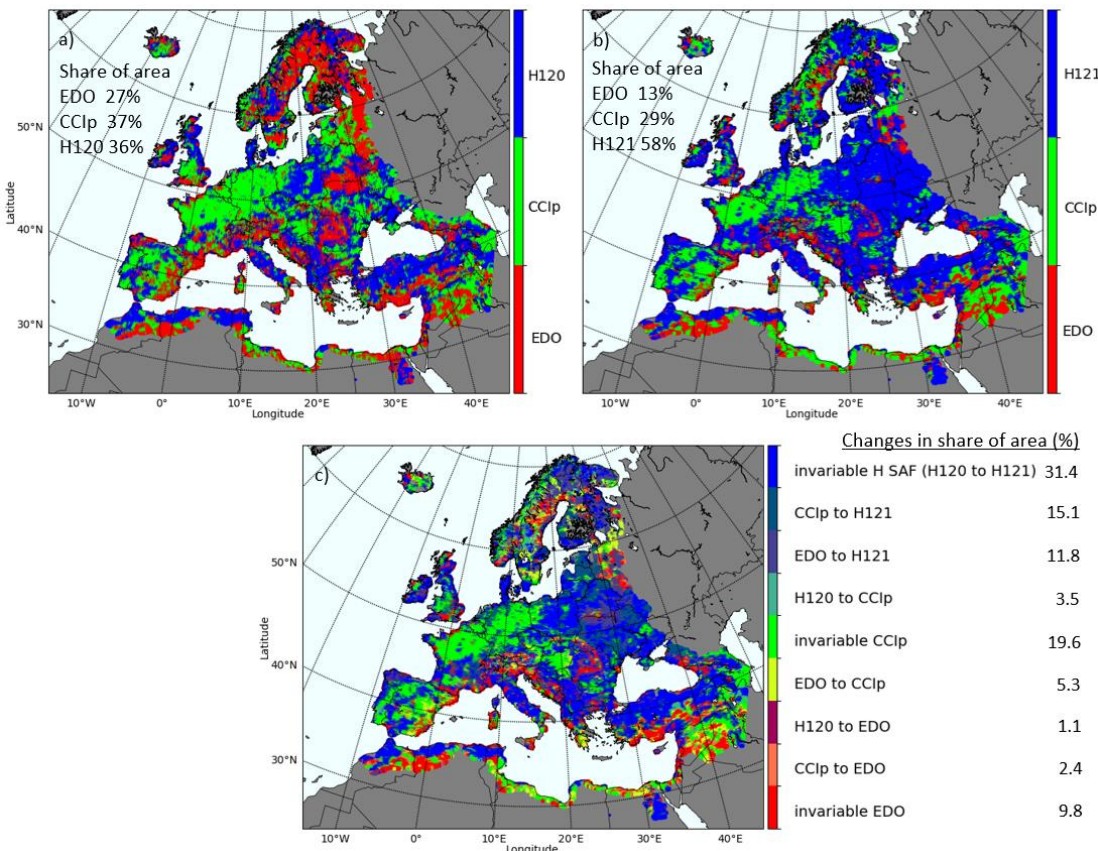

**Figure 6: Map of best performing SM product over Europe for a) the triplet H120, CCIp and EDO and b) the triplet H121, CCIp and EDO. c) Map identifying all changes of best product (from product x to product y) and the areas that stay invariably best estimated by each product with quantifications in % total area.**

## 4.3 Active, passive and model-based SM products against in-situ ISMN data

The consistency of RS, model-based and merged SM products considered in this study is compared to the data of the European networks of the ISMN for their coincident periods (Fig. 7). CCIp coherently agrees for most of the networks with its active counterpart, the H120 / H121 products, both in magnitude and spread of correlations despite being the product with the lowest overall R mean/median score at the ISMN (R $_{CCIp}$=0.47/0.51). H120 / H121 active SM datasets perform second in terms of overall R Pearson correlation (Fig. 7) among the ISMN (R $_{H120}$=0.46/0.53, R $_{H121}$=0.51/0.51). H120 is the SM product showing the widest spread of correlations in some of the evaluated ISMN such as BIEBRZA_S-1, GROW, GTK, XMS-CAT. The lowest correlation values are also seen in FMI, GTK, XMS-CAT networks. Apart from GROW network, prone to high uncertainty (Zappa et al., 2020), and XMS-CAT which length of series may be limited, FMI and GTK usually show poor correlation values with RS data (Kolassa et al., 2017; Ikonen et al., 2018) attributed to the dense boreal canopies challenging active and passive sensors (Petropoulos et al., 2015; Kerr et al., 2012).

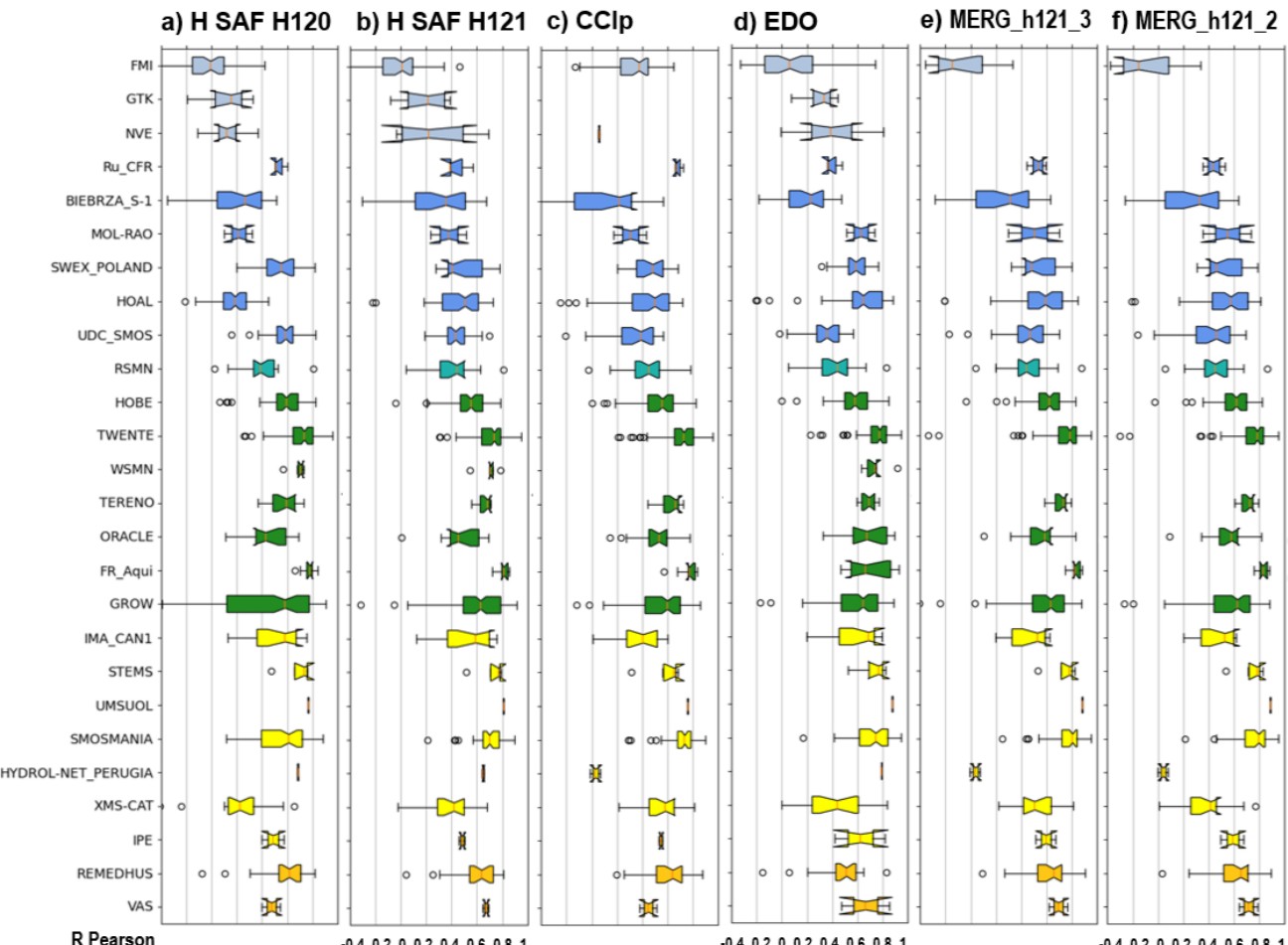

**Figure 7:** R Pearson coef. of a) H120, b) H121 c) CCIp, d) EDO, e) the merged MERG_h121_3 (combining H121, CCIp and EDO), and f) the merged MERG_h121_2 (combination of H121 and EDO) against in-situ SM data of the ISMN (names of networks in the leftmost column). The colour of the notched boxplots corresponds to Koppen Geiger climatic classes (gold colour=*B* climate class, yellow: *Cs*, Green: *Cf*, Blue: *Dfb*, Blue green: *Dfa*, Ice blue: *Dfc-E*) assuming all stations of each network have same climate. In the Y axis ISMN networks sorted from the northernmost to the southernmost in latitude within each group of climates, and climates sorted from colder to warmer.

The clearest differences in R score across all products occur precisely over boreal areas of *D-E* climate types (Fig. 8b). The rather high R scores of EDO (R $_{EDO}$=0.56/0.6) also show a downgrade in performance (Cammalleri et al., 2015) (Fig. 8) due to its better calibration for rain-dominated than for snow-dominated regimes (Salamon et al., 2019) and in line with the 'seasonal' R_TCA results commented in section 4.1.2 where the performance of EDO during the snow-covered period of winter (Dec 15-March 15) proved noticeably lower than that of RS products. However, the low performance of EDO compared to the other products over boreal areas is in fact better at the specific locations of the ISMN networks of Scandinavia (FMI, GTK, NVE) compared to the results when R Pearson or R_TCA scores are considered. This disagreement indicates more the need of additional networks in such a vast boreal area than actual better performance of model-based estimates over the remote sensing ones. The values of SM estimated by EDO agree to a great extent with those of H120 / H121 and CCIp, which exceed

globally, the mean/median scores of these RS products. Again, the apparent surpass of EDO scores at ISMNs over H120/H121 and CCIp (Fig. 8a) may respond more to the limited locations of the ISMN networks instead of to the actual superiority of the model-based estimates, because EDO's show wider spread of correlations than H121 and CCIp (Fig. 8b) not only over the snow-prone *D-E* climate types but also for *Cs* and *B* type climates. EDO shows the smallest correlation spread over the *Cf* climate type which is the closest to the ideal hydrological regime emulated by conceptual models. The 5x5km resolution of EDO compared to the coarse CCIp and H120/H121 may also play a role in the apparent better values of EDO than of CCIp and H120/H121 (Fig.8). The low performance of EDO over the UDC_SMOS network may be related to the anomalous SM and flood conditions experienced over the region of this network in the upper Danube during the period 2007-2011 (Wanders et al., 2014). EDO performs best at two local networks: UMSUOL and HYDRO-NET-PERUGIA. After all, both scale effects of the products and the representativity of the ISMN networks are related.

Furthermore, products may cover better or worse the range of SM variability shown by the ISMN stations affecting the overall score of each product. When the range of the notch surpasses the interquartile range of the boxplots in a network, which occurs in local-scale networks such as MOL-RAO, Ru_CFR or VAS, RS or model-based data might be unable to display the SM variability that local-scale networks can describe (Brocca et al., 2010a). EDO has the least prevalence of notched boxplots. Although the measuring technology plays a role because using less accurate techniques (e.g. GROW, BIEBRZA_S-1, XMS-CAT) tend to show also more spread at the ISMN data (Dorigo et al., 2021), other factors such as land cover may be more influential. The higher spread (Fig. 7) or lower correlation (Fig. 8) of CCIp over some networks of heterogeneous land cover or in the extremes of the range of soil moisture conditions of the ISMN networks (e.g. XMS-CAT or BIEBRZA, respectively), can be due to the coarser resolution of CCIp compared to H120 and H121 (Dorigo et al., 2010).

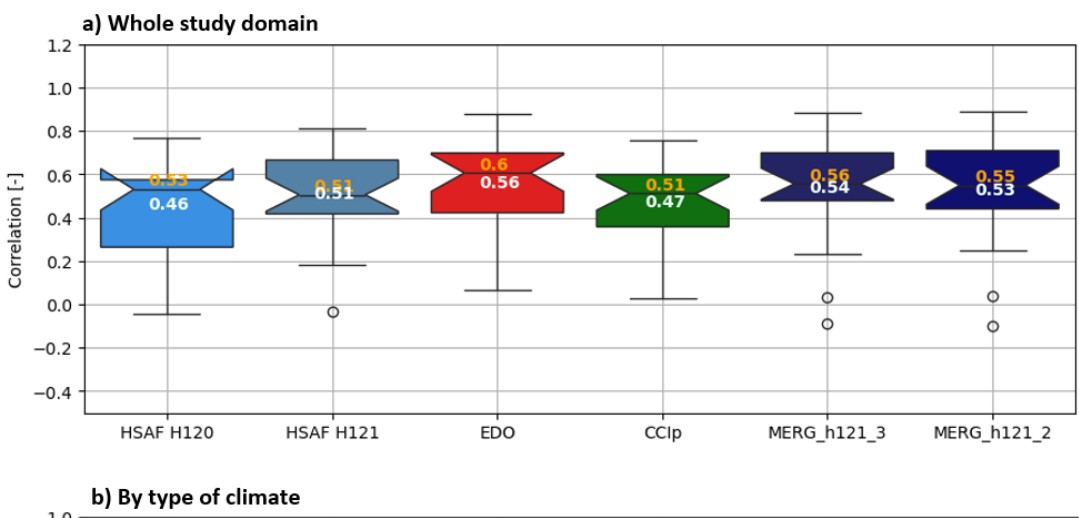

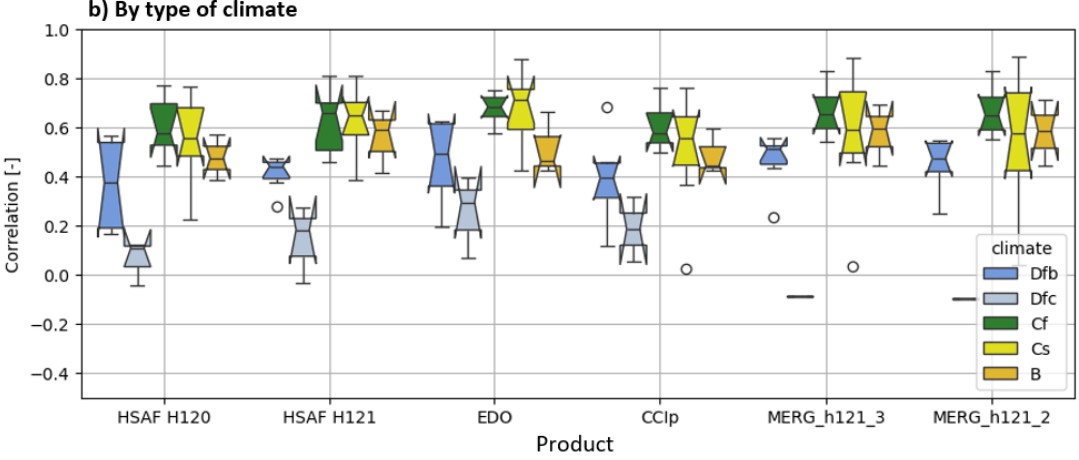

**Figure 8: a)** Boxplots illustrating the distribution of the R Pearson correlation coef. of the active RS SM product H120 and H121, the passive CCIp, the model-based EDO and the two suggested merged products with superimposed mean values in white and median values in yellow colour. **b)** Boxplots of the products for the subsets of the different climate types. The notches represent the confidence interval of the median and when surpassing the interquartile ranges indicate uncertainty, partly due to a small size of the samples.

The results of the merged products in Fig. 7 and 8 indicate that the weighted combination of techniques surpass the performance of H120 / H121 versions and CCIp (R $_{MERG\_h121\_3}$=0.54/0.56, R $_{MERG\_h121\_2}$=0.53/0.55). Only EDO shows higher scores thanks to its higher performance station by station and over the areas of climate *Dfc-E* that are also challenging for RS SM products (Fig. 7 networks in blue and ice-blue, and at sub-boxplots of types of climates at Fig. 8b). In the rest of climates MERG_h121_3 and MERG_h121_2 tend to reduce the spread of the scores at ISMN and slightly increase their value. Despite the lower tails of the merged products propagating from climate types such as *Cs, Dfc-E* with short, local or reduced number of series, their distribution of values is better than that of individual products, especially for the interquartile range (Q1-Q3 over the boxplots of Fig. 8a). Climate types that prevail across the continent such as *B, Cf* and *Dfb* are the most benefited, except for EDO, by the merging. Here, the merged products adopt a rather balanced weighting, but any other merging scheme favouring the best performing product in an area may notably enhance the performance of the merged products. Furthermore, the MERG_h121_2

(combining H121 and EDO) almost equals the results of the MERG_h121_3 product, which evidences that best performing products can be obtained even without using CCIp, emphasizing the possibility to obtain a merged product solely based on RS and modelling data available in near real-time.

## 4.4 Evaluating the trends on SM databases and discussing the implications

Results of the analysis of trends of the monthly anomalies in the period 2007-2022 of H120 / H121, CCI and EDO exhibit spatial and temporal contrasts (Fig. 9). There is partial agreement between the RS products CCIp and H120 / H121 (Fig. 9a, $c_1$ and $c_2$) not only in wet anomalies but also in a few drying areas. The relatively higher agreement between CCIp and H121 across the continent is due to the less extensive wet trend of H121 but is consistent with multiple validation studies in the area (Gruber et al., 2019; Preimesberger et al., 2020) and with reanalysis data (ERA5-Land, not shown here, Pierdicca et al., 2015a).

However, there is notable contrast between the drying trend of EDO (Cammalleri and Vogt, 2016) and the wet trend of H120 (Wagner et al., 2022) (Fig. 9b vs. b). The products have noticeable diverging trends (positive in EDO, negative in H120) but they still agree in the sign of areas where products concur in subtle trend which indicates relative spatial agreement despite not agreeing on the magnitude of the trends of the anomalies. Therefore, as EDO and H120 do not surpass CCIp range of trends,

they can be considered as the products depicting the lower (EDO) and upper (H120) range of trend characterization. They both additionally agree in extensive significance of the trends which suggest inherent issues of the products with trends, and subsequently, the products with lesser extent of significant trends are considered here of better performance in line with works indicating trends might be less widespread than expected (Almendra et al, 2022). The divergence between H120 and EDO and the different range of the significant trends between CCIp and EDO (Fig. 9) illustrate the need to refine products, which recent

versions (e.g. H121) seem to have accomplished.

The merged products show intermediate characteristics as results of combining H121 and EDO (MERG_h11_2) or also CCIp (MERG_h121_3). Their trends become more balanced in sign and lower in magnitude compared to their original products which also supports the use of merged products for specific applications addressing anomalies sensitive to drifts

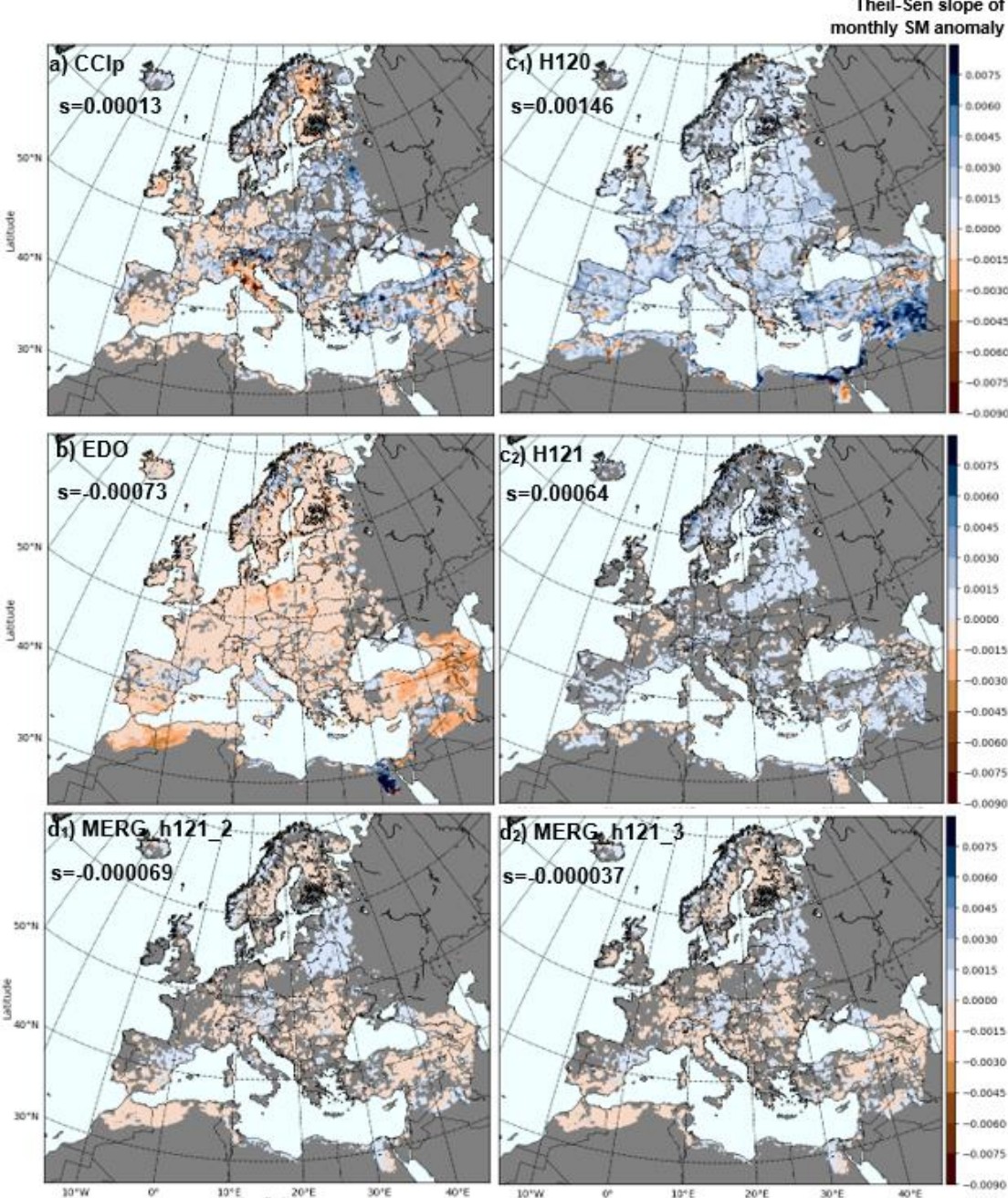

Figure 9: Maps of significant annual trends (Theil-Sen slopes) of the series of monthly SM anomalies indicated by the Mann-Kendall tests of a) CCIp, b) EDO, $c_1$) H120 and $c_2$) H121, and the merged products $d_1$) MERG_h121_2 and $d_2$) MERG_h121_3 for the period 2007-2022. Non-significant areas in the plain grey colour of the rest of continental areas outside of the domain of study. 'S' describes overall slope values.

Interestingly, both merged products (Figs. 9d$_1$ and d$_2$) agree to a great extent, which suggest that the role of CCIp might be already secondary when H121 and EDO become merged. Such agreement in between the two may also imply that the dominance of H121 as best product (Fig. 6) is due to the already consistent nature of H121 or alternatively that H121 and CCIp are both in great agreement. Though, by adopting H121 the significant areas formerly dominating EDO or H120 become no longer the norm and also reduced compared to CCIp (Fig. 9b vs. d$_1$ and d$_2$). Therefore, merged products compile the trendy areas with more consensus in sign, magnitude and location in between the products, which can be considered as a consistent depiction of the major trends.

The cause of the SM trend shown by EDO (Fig. 9b) may seem attributable to global warming origin based on what is widely accepted to be impact of climate change or based on previous reports using reanalysis and model-based studies (Samaniego et al., 2018; Li et al., 2020). The drying trend has been shown to prevail in EDO (Dorigo et al., 2012; Almendra et al., 2022) at least in southern latitudes (Cammalleri et al., 2016). However, many areas of low dry trend in EDO do not concur with the drying areas of H120 / H121 and CCIp (Fig. 9), especially in the SE or NE of the domain where mixed and wetting trends of H120 and CCIp have been described (Tuel and Eltahir, 2021, Saffioti et al., 2016), but partially agrees reports of SM sensitivity to temperature change using GLDAS (Gu et al., 2019). Therefore, it is still appropriate to contrast SM trends with those of related variables such as precipitation, evapotranspiration, temperature (Meng et al., 2018; Deng et al., 2019) or even with the response of vegetation (Liu et al., 2020; Lal et al., 2023). The temperature influence may suggest that EDO overexpresses SM trends due to sensitivity to meteorological forcing (Koster et al., 2009). The EDO series over the most temperature-driven climates (e.g. B-type, Fig. 10b$_1$) partially agree with that. However, since the temperature is a variable only indirectly influencing the water balance , for the case of a preliminary exploration of the correspondence between SM trends and auxiliary variables we prefer to refer to precipitation and evaporation trends as components of the water balance directly affecting SM. Significant ERA-Land precipitation and evaporation trends occur at fewer areas than those shown by SM products, therefore, the areas concurring in trends strictly circumscribe to areas with precipitation and evaporation trends. The southwestern part of Europe indicated by EDO does show a decreasing SM and precipitation for EDO and ERA5-Land, with an increasing trend in evaporation. In the SE and Central Europe there are diverging trends. This reduces areas of consistent trends between the SM and the auxiliary variables below the 8% of the total study area. This reinforces the idea of considering SM product trends more as indicator of SM product's consistency than of environmental change, which indeed would require specific methods to unveil causality between trends of SM and other variables.

That is why we interpret the EDO notable trends as inherent of the product. This possibility emphasizes the need to carefully revise implicit trends before use, particularly if climate change applications are intended. The successful upgrading of the active dataset from H120 to H121 version also illustrates the product's specific nature of trends, since product upgrades incorporating improvements on processing (subsurface backscattering: Wagner et al., 2022; vegetation: Vreugdenhil et al., 2016) and decreasing  their dependence from proxies (Dorigo et al., 2017; Madelon et al., 2021) have led to a significant reduction of SM trends between consecutive versions (e.g. here in between H120 and H121, in which the trend in SM reduces from the 66% of the total area in H120 to merely a 22% in H121).

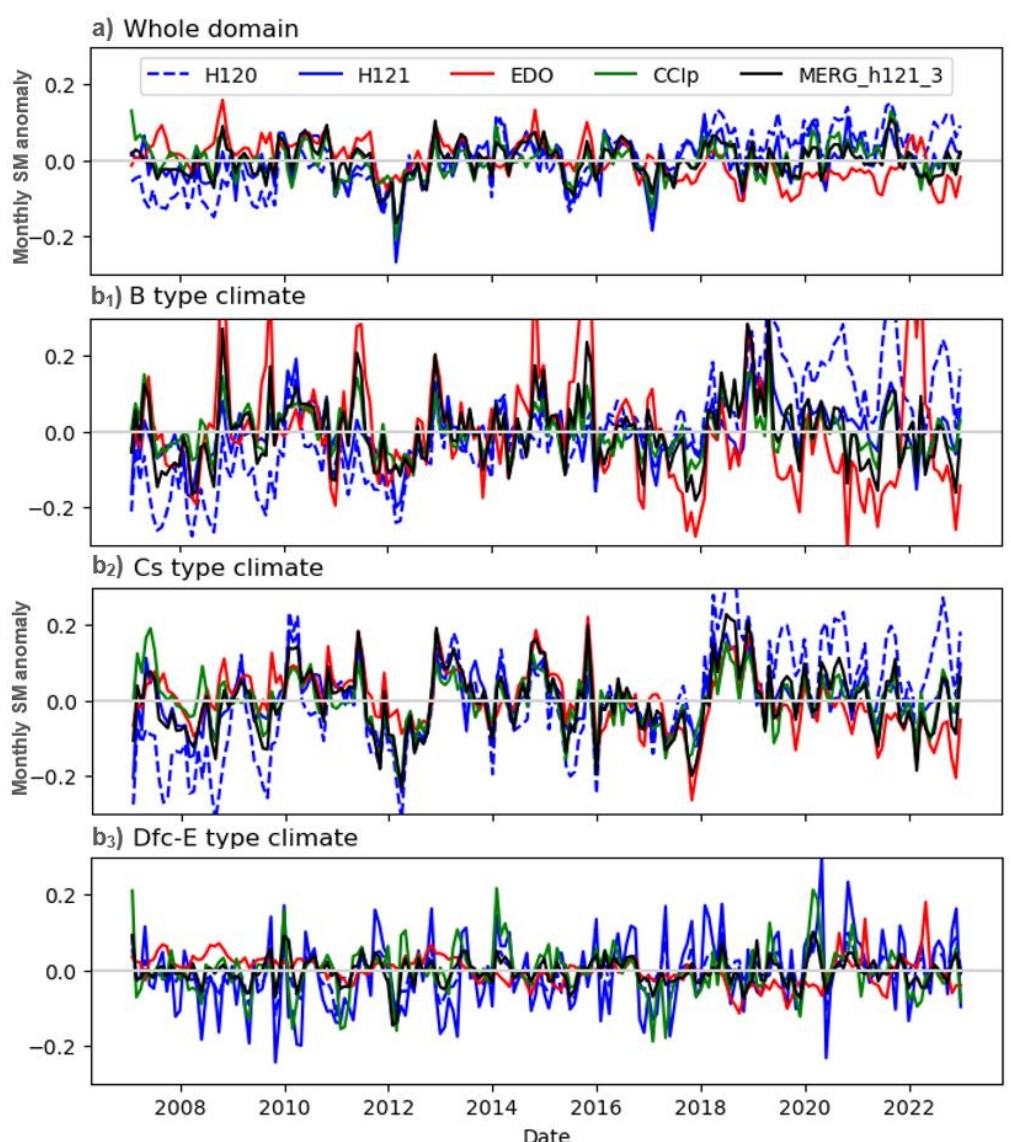

**Figure 10: Temporal trends of the RS (H120 / H121, CCIp), model-based (EDO) and merged SM products for the whole domain and period of study 2007-2022 of the monthly SM anomalies. (b) Temporal trends of the SM products of EDO (in green continuous line), H120 (in discontinuous blue line) and CCIp (in red line) in the study period 2007-2022, assessed by the main climatic classes of Europe b1) *B*, b2) *Cs*, b3) *Dfc-E*.**

The sequence of wet and dry spells displayed in the time series of Fig. 10a give the temporal detail of the spatial patterns shown on Fig. 9. The temporal divergence of the trends between EDO and H120 is visible in time series of the whole domain (Fig. 10a) and in the ones of the climate types (Fig. 10b1-b3), particularly in the last years (in Fig. 9 H120 show significant

trends in the 66% of the area, of which 61% positive and 5% negative, while EDO significant trends cover the 65% of the study area, of which 5% positive and 60% negative). Areas of *B* and *Cs* climates seem the most affected, followed by *Dsb*, *Dfb*, *Cfb*, *Dfa*, with *Dfc-E* as the least affected, in line with the temperature gradient. An example is the contrast between the former patches of SM increase identified in the period 1988-2010/2015 (Dorigo et al., 2012; Albergel et al., 2013a; Liu et al., 2019), Piles et al., 2019), and the decline reported recently (Skulovich et al., 2023) or in the past (Deng et al., 2019). When the joint trends of SM and ERA5-Land Precipitation and Evaporation are considered, the remaining patterns of trends concur more in between products than when only the SM trends are considered. Less than 5% of the total area shows joint trends among all SM products, but the agreement in between them in those regions, even with deferring sign of trends, is much higher than when considering only the SM trends (e.g. SW Europe and NW North Africa, pockets of Central Europe, Eastern Europe and Caucasus, and certain exposed areas of Scandinavia. These results support the consistency in between products while highlighting the minimal impact of auxiliary variables in the residual trends of the SM products. The best merged product, MERG_h121_3 show in Fig. 10 has the advantage of an overly balanced trend. Combining the diverging trends of H120 and EDO may have neutralized the trend in MERG_h121_3 and foster its insensitivity to the climate (Fig. 10b$_1$-b$_3$). A balanced spatial and temporal consistency of the merged product able to counteract the diverging trends of the individual products is preferable towards operational monitoring but it may also obscure the interpretation of the causes behind the diverging trends of its components. However, when products with diverging trends counteract, such as between EDO and H120, MERG_h121_2 agrees with the extension and significance of the trendy areas of CCIp. Thus, merged products can provide better temporal stability than the products used in their combination which is of benefit not only for operational monitoring but for long-term change analysis as well.

**5 Conclusions**

The evaluation of the complex soil moisture processes across all environmental conditions is challenging. The increasing capabilities of remote sensing and modelling datasets have eased the systematic monitoring of soil moisture, but their limitations require attention to prevent misleading interpretations. Well-known global soil moisture products such as the remote sensing active H SAF ASCAT-SSM-CDR-12.5km-v7 (H120) and v8 (H121), the passive ESA CCIp and the model-based EDO represent the three main distinct types of data suitable for soil moisture monitoring at the continental scale, each with specific strengths and weaknesses. This study illustrates that no single products excel universally, but that evaluating their spatial and temporal consistency as well as their uncertainties, particularly regarding residual trends, provides benefits, from which the strategic combination of data can enhance the operational monitoring of soil moisture across a wider range of conditions than when using individual products alone.

The correspondence of EDO, CCIp and H120 or H121 shown in pairwise correlation and TCA proved notably consistent in between products for most regions across Europe. The remote sensing datasets H120 and H121 and CCIp can provide equal, or better soil moisture estimates across most of the continent than EDO, prevailing CCIp over temperate oceanic (*C* type of

Köppen-Geiger classification) and H120 / H121 over temperate continental climates (*D* type). Conversely EDO, as a model-based type of data is still best at characterizing SM over areas where RS products experience uncertainties due to rough terrain or subsurface scattering. Known the distinct capabilities and differences of the products but also their synergies and increasing accuracy, the merging of these three types of SM data based on their performance scores provides equal or better soil moisture characterization than individual datasets alone, partly due to compensating biases and trends, which allows the soil moisture

monitoring across a wider range of conditions while optimizing the use of data.

Products depict a notably reliable characterization of soil moisture also against the in-situ data of the International Soil Moisture Network across most climates except for boreal ones (*Dfc-E* type) and despite the uncertainties due to network's particularities. Nonetheless, the passive RS CCIp and the model-based EDO, despite their spatial agreement and adequate comparability to in-situ data, still exhibit residual trends that obscure the interpretation of authentic soil moisture tendencies,

either in location, magnitude and significance. CCIp, while of balanced distribution between positive and negative trends over the continent, tends to display spots of excessive trend over small areas which disagree in sign and magnitude with other products. Conversely, EDO exhibits a tendency to show extensive areas of significant negative trends. The known positive trends of H120 have been corrected in H121 improving the sign magnitude, extent and significance of the trend portrayal. Yet, the overall trend agreement among products remains moderate because it implicitly contains product uncertainties not related

to real SM change.

In this way, the combination of datasets for operational monitoring of soil moisture maximizes the consistency of the estimates across environments and provides the most frequent revisit times with more balanced temporal stability, which benefits the applicability of soil moisture data for multiple applications that may be limited by the specific restrictions of each product.

Therefore, the merging, by understanding the capabilities and especially the limitations of SM products, fully exploit the value of the data accordingly to their preeminent characteristics. This aim is in line with the increasing demand of combined datasets that maximize the information provided by multiple sources of data while simplifying the effort on their evaluation and interpretation. Consequently, the manuscript demonstrates this possibility highlighting the benefit of this approach for enhanced operational monitoring of soil moisture.

**Authors contribution**

LB, DB, GF, SP designed the study; JG, LB, SC, PF, SH, PS and GF conducted the analyses; JG, LB, PF, SH, SC, GF, DB, PS and SP interpreted the analyses; JG wrote the manuscript draft; JG, LB, PF, SH, SC, DB, GF, SP, HM, NR, LC reviewed and edited the manuscript.

## Data availability

All raw data can be provided by the corresponding authors upon request. Additional associated datasets of the monthly z-score anomalies of the EUMETSAT H SAF H120 dataset have been published in Zenodo (https://zenodo.org/records/14706534) in the framework of the OEMC project funding this study (https://zenodo.org/communities/oemc-project/records?q=&l=list&p=1&s=10&sort=newest). Updates of the dataset will be soon available for EUMETSAT H SAF H121.

## Competing interests

The authors declare that they have no conflict of interest.

## Acknowledgements

The authors acknowledge funding from the European Union "Open Earth Monitor Cyberinfrastructure" project (grant agreement No. 101059548), the EUMETSAT "Satellite Application Facility on Support to Operational Hydrology and Water Management (H SAF) CDOP 4" project (grant no. SAF/H SAF/CDOP4/AGR/01) and the Italian Department of Civil Protection.

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
