# Peer review of "Soil moisture products consistency for operational drought monitoring in Europe"

_Hydrology and Earth System Sciences, 2024_

## Referee Comment (RC1)

**Review: Soil moisture products consistency for operational drought monitoring in Europe**

October 2, 2024

**1 Overall summary of the main contribution of the paper**

The paper is interesting and gives an insight into different soil moisture products leaned towards remote sensing products (2 HSAF versions and ESA CCI passive). They are complemented with one model dataset from Lisflood (EDO). All data are evaluated using in situ data from ISMN and the triple collocation method. The aforementioned products are then merged into 2 combined products and evaluated against ISMN. Finally. the authors make a trend analysis with the dataset mentioned.

The conclusions of this study are mainly well informed by the analysis and plots

**2 Appropriateness for the journal and relevancy**

The study is in the scope of HESS and fits in very well. The topic of soil moisture assessment and monitoring highly relevant.

**3 Major items should be addressed in revision**

- The authors should make clearer what is the novelty and the research gap addressed by their study.

- I suggest to change the title by removing the term drought as the paper is not reffering to droughts at all.

- How are quality assurance scores (ESA CCI, ISMN) treated within this study?

- L212 How do the datasets used fulfil these assumptions?

- L297ff The low scores of EDO in Northern Europe may arise from low performance but good agrrement of the other 2 RS products. As you mention before they are known to worse in forested and icy regions. Please elaborate a little in the discussion and add in situ observations as reference to get a better picture.

- L361 Please check the time period of XMS-CAT to underpin your analysis.

- L228I couldn't find the mentioned method in Braocca 2010b

- L3956 In the rest, ... - Compared to EDO the spread of B and C type climates seems not to be reduced in the combined products.

**4   Minor corrections**

- L76 please rephrase numerical model into process based or conceptual model.

- Fig. 1 Labels a and b missing.

- L171 Web address of ISMN is outdated. Please update with https://ismn.earth.

- Fig. 2 A distinction into further classes like forest, urban etc would be useful.

- L190 Please rephrase (The daily values ...)

- L197 I guess $SM_{sat}$ shall be $m_s$

-

- I suggest to add a table detailing the different spatial and temporal representation/resolution of the data source.

- Please review the use of hyphens instead of a comma when you refer to two panels in a figure, e.g, L260 (Fig. 3b-3e), L261...

- L229 $R\_TCA_{HSAF}$ should be $SM_{HSAF}$

- L270 caption of Figure 3: Map of temporal R-Pearson correlation instead of spatial.

- L318 English grammar, please rephrase (Hence, ...).

- Fig. 7: Please add an axis label to all x axes.

- L372 Please rephrase, sentence is hard to understand.

- L399 I am not quite sure about which dashed lines you are talking about.

- L400 Dfb instead Db

- L 414 Please rephrase, sentence not understandable (While the products ...)

---

## Author Response (AR2)

**Reply to 3rd and 4ᵗʰ reviewer on the manuscript of title:**
**"Soil Moisture consistency for operational drought monitoring"**

We thank the 3ʳᵈ reviewer Vagner Ferreira and an anonymous fourth reviewer for his insightful comments to improve the manuscript and its significance for providing guidance for operational users. In the following lines we answer both their major and minor comments.

**Major items**

- Page 9: The authors state they use a T-value of 10 days for the SWI calculation, which seems arbitrary. Have the authors carried out any sensitivity analysis performed to determine the optimal T-value for different regions? Different soil types and climate regimes may require different characteristic time lengths.

  The use of the T-value for the exponential filter used to calculate SWI is substantiated by specific studies that after having explored the optimal value, based on validation against in situ data, indicate that the T=10 days  Albergel et al., 2010, Brocca et al., 2010 (coauthor of this manuscript)) is the optimal value. The T =10 days value is reported to provide an optimal balance between noise reduction and signal preservation for the uppermost layer. The use of non-constant T value across the areas of interest (depending on the climate, soil type) does not significantly increase the performance of results of articles using some of the products of this manuscript (i.e. ASCAT) (Brocca et al., 2011; Brocca et al., 2010).

  Nonetheless, since the continuation of this work is devoted to infer the impact of multiple physical factors in the differing capabilities of the products, particularly in relation to their temporal resolution (e.g. snow and water fraction for thawing and flooding circumstances), there are ongoing analysis that will analyze different T-values, and that for the sake of conciseness are not included in this manuscript.

- Page 10: The triple collocation analysis (TCA) assumes error independence between products, but this assumption may be violated, especially between remote sensing products that share similar physical principles or between products that may use similar auxiliary data. The authors should discuss potential violations of TCA assumptions more carefully.

  We recall some of the content prepared to improve the manuscript in the revision to explain the reasons why we considered TCA assumptions have been handle with

care so that results in this manuscript are valid:

The TCA model assumes (1) linearity of SM retrievals, (2) stationarity of signal and (3) independence of errors (Gruber et al., 2016; Massari et al., 2017; Filippucci et al., 2021).

For the first case (1), we selected SM products for triplets based on their reported independence. The LISFLOOD model is not used for the processing or validation of either the passive and active RS products that complete the product, nor for the curation of the ISMN data. The passive subset of CCI included here ('CCIp'), does not contain active remote sensing data (i.e. ASCAT). In this way, independence of the datasets ensures non-linearity between the three products of TCA.

The major source of non-linearity can be due to non-stationarity (2). The upgraded version of ASCAT active RS SM (H121) filters out the trend existing in earlier versions. CCIp trend has been recognized as consistent in multiple works and can be considered only marginal due to its dedicated blending methodology. Residual trends are principal for long term analysis but are negligible for the short-term scope of operational monitoring in this text, and so are they for the TCA results exposed.

We additionally evaluated the stationarity of the variance by comparing metrics (R_TCA, 'VAR'iance of the signal, 'SENSi'tivity of the signal and logarithmic signal to noise ratio 'logSNR') between two sub-periods of distinct trend magnitude: 2007–2015 (low trend) and 2015–2022 (high trend). Results show no significant differences between periods and in between 'trendy' (e.g. ASCAT H120) vs. 'stabilized' products (e.g. H121), confirming that neither the magnitude of the trend nor the period selected compromise TCA validity (Section 4.4).

Regarding the assumption of the independence of residuals, overall, the number of triplets available in the period of study is far bigger (up to 536) than the 100 - threshold estimated by Scipal et al. (2008) to ensure the independence and no bias of the residual errors (Fig. 1 of this response). Even the regions prone to snow cover present enough coverage (far beyond the 100 values of the threshold) to guarantee no downgrade in the reliability of the TCA model due to lack of data representativity.

About the assumed linearity between the signal and the errors (3), values of the logSNR in decibels stay within ranges of linearity of the relationship between

logSNR and R2 (Mean around 5-7 dB). Debates on the linearity as reported in (Gruber et al., 2015) indicate that despite the bias introduced by this assumption there is no clear alternative to the consideration of linear models to the relationship between the model and the data because other methods also introduce bias through matching, scaling and application of polynomials.

[Figure]

*Fig. 1: Coverage in number of valid triplets for the analysis of the three products ASCAT H120/H121, CCIp and EDO across Europe in the period 2007-2022.*

- Page 11: The merging methodology appears to be primarily based on TCA scores (Comment 2). However, this approach may propagate biases in regions where all products perform poorly. A more sophisticated merging approach that considers the physical basis of errors would improve the quality of the analysis.

We appreciate the reviewer's concerns regarding the potential propagation of errors when merging soil moisture products based on their Triple Collocation Analysis (TCA) scores, particularly in regions where individual products exhibit lower performance. We acknowledge that low performances may still occur in the results based on merging by TCA-score weights, but as section 3.4.3. indicates, the

merging favors the better-performing products at each pixel, including over the error-prone areas. However, we would like to underline that for the focus of this manuscript at the continental scale, these regional limitations, that are nonetheless specifically warned, do not diminish the worth of highlighting the benefit of merging highly performant products in the majority of the areas of the continent. In this way, the weighted but distributed approach may counterweigh good estimates in the presence of low scores but still provides results dominated by the best product across the domain. While more complex methods may achieve better results, they often require tricky tuning, assumptions that introduce their own uncertainties or validation schemes with ancillary data. Until we get ready the insights from other factors, TCA-weighted merging remains a transparent, assumption-lean, and computationally efficient alternative, that has been acknowledged as reliable ( Yilmaz and Crow, 2014; Gruber et al., 2017; Chen et al., 2019) and that serves well for the purposes of this manuscript to illustrate how merging (regardless of the method) can benefit from combining the virtues of each product.

Comparison has been done for the merging method whose results were included in the manuscript against a modified version of the merging method considering the exclusion of the products that do not achieve tolerable R_TCA scores (e.g. R_TCA<0.4) in pixels over areas of difficult SM estimates such as boreal regions. The results of this 'exclusion' method do offer better overall results by including only those remaining areas with more than tolerable R_TCA scores, but with the disadvantage of greatly reducing the coverage in area of the estimates (more than 40% of the coverage of results over even areas of only intermittent snow conditions (see the R_TCA scores of EDO during the snow season of most of Central and Eastern Europe in the Figure 2 in the next page). We consider that the notable reduction of the coverage at the expense of better overall performance jeopardizes the aim of maximizing the operational suitability of the products, which is the goal of the study. Furthermore, since the performance also varies seasonally, it would be necessary to apply also temporal variations in the weighting of the merging, something of difficult definition taking into consideration the weights are precisely defined based on calculating the R_TCA per pixel along the time series.

Nonetheless, we are aiming to adopt more flexible merging methods to deal with these challenging considerations of the sensitivity of merging to the inputs and to the definition of the combination in order to prevent the buildup of uncertainty in the merging process.  We acknowledge the relevance of tackling these matters,

especially in focus of particularly challenging regions (e.g. snow-prone areas where most products perform poorly) that are of interest. Consequently, since the evaluation of the characteristics of the input datasets has been extensively described in this manuscript, the continuation of this study will primarily focus on the comparison of merging techniques for an optimal combination of the input data.

- Page 12-13: The negative correlations between products in snow-dominated regions are concerning, which has been attributed to the known difficulties with EDO-Lisflood in snow regions. However, this raises questions about the reliability of any of these products for drought monitoring in such regions. This limitation should be more prominently acknowledged.

  We have compared TCA results of the period of maximum snow cover likelihood (15 of December to 15 of March) compared to the TCA results of the snow-free season (15 of May to 15 of Oct) to better illustrate the impact of snow cover in the suitability of each product. The periods were defined based on ERA5L avg. Snow-cover. The Snow-covered period ('SP' from 15th of December to the 15th of March), displays the clear struggle of EDO product over lands that experience transient snow cover (yellow colors from Germany eastward). SM estimates from this product over these areas primarily formed by *Dfb* climate type, may be particularly unreliable during the period which is relevant to explain the gradients west-east obtained for the results of the entire year. Interestingly, ASCAT estimates remain reliable even during this period, which may suggest that despite the limitations of ASCAT over permanent snow-cover, ASCAT, followed by CCIp outperforms EDO on characterizing transitions of snow cover (the majority of the yellow area of *Dfb* climate type does not show permanent winter snow cover but episodic (intermitent, non-permanent along the whole winter season).

R_TCA only SP (15 dec to 15 March)

[Figure]

*Fig. 2: R_TCA scores of the two triplets considered (with the old and new versions of H SAF product) over the snow-prevalent season.*

R_TCA only NSP (15 May to 15 Oct) (New H121 v2)

[Figure]

*Fig. 3: R_TCA scores of the two triplets considered (with the old and new versions of H SAF product) over the snow-free season.*

The snow-free period ('NSP' from 15th of May to the 15th of October) displayed above suggests a balanced performance of the three products across most areas of the

continent, with the comparatively better performance of ASCAT over the other two products, especially after the update from version H120 to H121, whose increase was already visible in Fig. 4 of the manuscript. Nonetheless, the R_TCA values in the Finno-Scandinavian region remain comparatively lower than those of other continental areas with the exception to the regions also affected by specific sensitivity of each product, which indeed leads to raise awareness about its reliability over the boreal belt, which has been further expanded in Lines 388-390, L401-404, L483-485, L618-619.

- Page 15: The authors suggest that the west-east gradient in product performance may be related to SM regimes, but this connection is not explored further. So, it would be interesting to see a more detailed analysis of how different SM regimes affect product performance would support this claim.

We indicated before that the influence of physical factors is being explored, in particular in relation to the distinct soil moisture regimes determined by the climatic type (among other factors). Our preliminary results from Intensity-Frequency (I-F) curves of SM change along the whole time series and across climates indicate that there are significant differences in the frequency and intensity of soil moisture change for both rewetting and dry down conditions depending on the climatic areas, specially across the West-East gradient in between oceanic (Cfb) and continental climates (DS), or in between them and the Mediterranean climate (Bs).

The oceanic climate type (the west part of the gradient, first figure below) shows the broader range of suitable applicability of the products compared to those climates in the east (second figure below) and south of the continent (third figure below). This meaning that the SM products can be used with greater confidence even for the upper percentiles of SM change values (more extreme SM change values) while results in the east and south climates of the continent become less reliable with lower changes in SM (i.e. for a given intensity of SM change the return period becomes more different between products, as seen in the differences in color of the maps of the upper row of subplots of the figures below). Thus, the accurate range of use of the products across climates differs according to the differing accurate range identified from the intensity-frequency (return period) curves and plots.

To some extent, this outcome suggests current SM products perform best in the relatively progressive SM changes characteristic of Oceanic SM regimes compare to

those of Continental and Mediterranean characteristics experiencing sharper SM changes due to the influence of sudden meltdowns or torrential rains, respectively.

[Figure]

Fig. 4a: Percentiles of Intensity-Frequency (I-F) characteristics of the positive SM changes (delta SWI) in the temperate climate and threshold of concurrence (return period) of the SM products on the range of I-F values.

[Figure]

Fig. 4a: Percentiles of Intensity-Frequency (I-F) characteristics of the positive SM changes (delta SWI) in the continental climate and threshold of concurrence (return period) of the SM products on the range of I-F values.

[Figure]

*Fig. 4a: Percentiles of Intensity-Frequency (I-F) characteristics of the positive SM changes (delta SWI) in the Mediterranean semiarid climate and threshold of concurrence (return period) of the SM products on the range of I-F values.*

- Page 21-23: The interpretation of diverging trends between products is problematic. For example, if EDO shows drying trends while H120 shows wetting trends, this fundamental disagreement raises serious questions about using either product for climate change studies. The authors acknowledge this issue but should provide clearer guidance on how to interpret these conflicting signals.

The diverging trends between products is the reality we illustrate with this study. Multiple reliable soil moisture datasets indeed present diverging trends. Some of the residual trends may be related to internal characteristics of each product. Thanks to the permanent revision of the products some of these internal issues become solved and decrease the residual trend (e.g. HSAF update from H120 to H121 greatly diminishes the residual trend of the product). The study does indicate that the residual trends of the remote sensing products (CCIp, H SAF H121) present the less area of significance trends, better range of trends and better agreement in between them than model-based products (Lines 543-552). ERA5L cannot be used as reference of trends because EDO partly ingests its data, and to some extent also reproduces the patterns, and residual trend of ERA5L (although of enhanced magnitude in EDO's case). However, the trends of ERA5L may have their own level of uncertainty for diverse reasons (e.g. model intervention in the reanalysis) not only for soil moisture, but also for commonly observed variables (e.g. due to data

regularization). The manuscript further emphasizes that the existence of these trends recommends careful application of the data (L21-23, L127-128 extended in Lines 36-38, Line 585-87) by understanding their uncertainties and their possible impact on the outcomes of the analysis, which, of course, would be of major relevance for climate change studies.

Nonetheless, despite of the impossibility to further extend this manuscript with a section focused on the correspondence of trends with possible drivers of SM change such as the well reported changes in temperature, we provide a simple explorative correlation analysis of the trends of SM products in relation to the trends of two proxies of the water balance: total precipitation and total evaporation from ERA5-Land (ERA5L). The analysis circumscribes to exploring correlation since causality may require dedicated studies beyond the purpose of this manuscript.

Precipitation trends from ERA5L primarily show regional patterns of limited area (barely a 14% of the study area shows significant precipitation trends, with a 5% of positive trend limited to boreal and Mediterranean areas, and negative trends up to 9% over areas of central Europe, south Mediterranean and the Caucasus, in all cases of low magnitude. In this way, given the SM products tend to display a much larger extent of the study area affected by trends (H120: 66%, of which 61% positive and 5% negative; EDO: 65%, of which 5% positive and 60% negative, H121: 23%, of which 19% positive and 4% negative), the joint patterns are mostly defined by the few areas affected of precipitation and evaporation trends. However, since the signs of the trends can concur or not, there can be four classes of trends: coinciding positive trend (in blue), coinciding negative trends (in red), positive SM and negative proxy trend (orange) and negative SM with positive proxy trend (green).

To be concise, we show below only selected cases from all the SM products evaluated. The first one is that of the EDO model-based product, selected to illustrate how having vast areas of trends in SM products do not automatically imply broad agreement with proxies. While for precipitation and SM there are multiple areas of coinciding negative trends, for evaporation the results are more balanced in between coinciding and differing signs of the trends, which casts doubt on the physical meaning of the correlation of SM trends with those of the proxy.

EDO SM (left) and ERA5L (right) Precipitation significant trends 2007-2022

[Figure]

Joint EDO SM - ERA5L Precipitation significant trends 2007-2022 (Theil-Sen slopes)

[Figure]

*Fig. 5a: (upper left) Trends in EDO SM product, (upper right (trend in ERA5-Land Total precipitation) and (bottom) joint concurrence of trends between the above SM and Precipitation products.*

EDO SM (left) and ERA5L (right) Evaporation significant trends 2007-2022

[Figure]

Joint EDO SM - ERA5L Evaporation significant trends 2007-2022 (Theil-Sen slopes)

[Figure]

*Fig. 5b: (upper left) Trends in EDO SM product, (upper right (trend in ERA5-Land Total evaporation) and (bottom) joint concurrence of trends between the above SM and Evaporation products.*

In the second case, we provide the joint trends with precipitation and evaporation of the HSAF products: the H120 recently updated to version H121. In this way we can see the notable decrease in trends from H120 to H121, and how this upgrade limits the areas indicating concurrence of the trends. In H120, despite the general upward trend in SM, the areas with significant and noticeable joint SM-precipitation negative trends point again to areas of central Europe, south Mediterranean and the Caucasus, in full concurrence with EDO results. For evaporation, despite the class of the sign concurrence differs with EDO, the same trends areas are indicated, which highlights the consistency in between SM products (the proxy is common).

H120 SM (left) and ERA5L (right) Precipitation significant trends 2007-2022

[Figure]

Joint H120 SM - ERA5L Precipitation significant trends 2007-2022 (Theil-Sen slopes)

[Figure]

*Fig. 6a: (upper left) Trends in H120 SM product, (upper right (trend in ERA5-Land Total precipitation) and (bottom) joint concurrence of trends between the above SM and Precipitation products.*

H120 SM (left) and ERA5L (right) Evaporation significant trends 2007-2022

[Figure]

Joint H120 SM - ERA5L Evaporation significant trends 2007-2022 (Theil-Sen slopes)

[Figure]

*Fig. 6b: (upper left) Trends in EDO SM product, (upper right (trend in ERA5-Land Total evaporation) and (bottom) joint concurrence of trends between the above SM and Evaporation products.*

The H121 upgraded version of HSAF SM product illustrates the great reduction on the extent of the areas affected by trends, while confirming the trend in those areas where the magnitude of trend was most prominent in the previous version. By doing so, there is an obvious reduction in the areas affected by joint trends, still over the areas previously identified by the other products (including CCIp, not shown), both for precipitation and evaporation. It can be emphasized that by just the upgrade of this ASCAT HSAF product from H120 to H121, the areas affected by joint trends reduce from the 14% to the 3% (precipitation) and 18% and 5% (evaporation). This

reduction may indicate the secondary relevance of trend analysis for the purpose of change analysis (and subsequently of the interaction between SM and drivers of its change) compared to the relevance of trend analysis for the aim of evaluating the evolving capabilities of SM products, particularly when subject to updates in the internal characteristics of the products (changes in the parametrization of model-based ones like EDO, or in the signal processing algorithms of the RS SM ones).

H121 SM (left) and ERA5L (right) Precipitation significant trends 2007-2022

[Figure]

Joint H121 SM - ERA5L Precipitation significant trends 2007-2022 (Theil-Sen slopes)

[Figure]

*Fig. 7a: (upper left) Trends in H121 SM product, (upper right (trend in ERA5-Land Total precipitation) and (bottom) joint concurrence of trends between the above SM and Precipitation products.*

H121 SM (left) and ERA5L (right) Evaporation significant trends 2007-2022

[Figure]

Joint H121 SM - ERA5L Evaporation significant trends 2007-2022 (Theil-Sen slopes)

[Figure]

*Fig. 7b: (upper left) Trends in H121 SM product, (upper right (trend in ERA5-Land Total evaporation) and (bottom) joint concurrence of trends between the above SM and Evaporation products.*

Consequently, since the purpose of this manuscript is primarily devoted to the intercomparison of SM products in terms of consistency, we suggest to keep the above analysis of joint concurrence of trends for the studies that being interested in long term evolution of SM devote to specific and more precise techniques of SM analysis and causality. Therefore, while comments have been included in the text

about the main messages of the analysis (L585-605, 615-625), these results at maximum may get included as supportive material in the manuscript.

**Minor comments:**

7. The abstract could present more clear messages about the practical implications for drought monitoring applications.

We thank the reviewer for identifying the need to expand the implications of the article in the abstract. The following lines have been added to the last paragraph of the abstract (L36-38):

*"We obtained that even these popular datasets are subject to patches of spatial inconsistency and residual trends when compared to the in-situ data from the International Soil Moisture Network (ISMN). These uncertainties have minimal impact on drought monitoring in most of Europe, except in snow-prone regions and for the assessment of long-term soil moisture trends used to design climate adaptation policies. "*

Nonetheless, in view of the great complementarity shown by the active and passive remote sensing and the modelled SM estimates, two merged products are proposed and tested against in-situ data. Results indicate that combining H SAF ASCAT, CCIp and EDO equals or surpasses the spatial and temporal consistency of the individual SM products alone, even when only the near-real-time products of H SAF ASCAT and EDO are combined. *"Thus, merging remote sensing and modelled SM products enhances spatial consistency, resolution, temporal coverage, and near-real-time capabilities for better European-scale drought monitoring, strengthening the early warning and risk management systems devoted to improving societal and environmental resilience." (L47-51)*

8. The conclusion section repeats much of the discussion and it could provide more clear recommendations for operational users.

*The conclusion has been carefully revised to better convey the benefits of the approach for operational users. (L636-678)*

9. Page 9: The use of flags from ESA CCIp as a mask for other products requires justification. Do these flags accurately represent data quality issues for other products?

We understand the reviewer's concerns regarding our use of CCI flags over other products (ASCAT H120 and H121, and EDO). However, we selected CCIp flags because they are the most sensitive and restrictive for snow detection. Specifically, ESA CCIp assesses snow

and frozen soil using both temperature and freeze-thaw conditions (via Ku-, K-, and Ka-band retrievals, as noted in the CCIp ATDB guide). This criterion of applying CCI flags to eliminate areas seasonally affected by snow cover to all SM products in boreal regions ensures consistency in the filtered areas.

It was indicated in the manuscript not in page 9 but in lines L256-259: *"The different product additionally provides metrics of the error characteristics to identify the areas and periods affected by relevant impactful factors such as snow cover. The flag scheme of ESA CCIp exemplifies the detailed procedures devoted to distinguishing when the data is subject to further filtering. Consequently, considering the notable importance of the snow cover factor in our analysis over the boreal and mountainous areas of Europe, we have applied the flags of ESA CCIp as mask to the data coverage of the other products of the analysis over the snow-prone regions. "* However, we have included also the text: *"Specifically, ESA CCIp assesses snow and frozen soil using both temperature and freeze-thaw conditions (via Ku-, K-, and Ka-band retrievals, as noted in the CCIp ATDB guide)"* within those lines.

10. Page 15: The statement that "none of the products alone can fully characterize SM across Europe with the same accuracy" is important but should be better quantified. What are the accuracy thresholds for different applications?

There are still no accuracy thresholds at all for the application of different products aiming to cover different applications. Acceptable overall values of error variance in the estimates are often assumed as appropriate below 0.04m3/m3 for calibration /validation of remote sensing products against in situ data, but considering estimates using only multiple types of remote sensing sources (e.g. with CCIp and H SAF H120 or H121) as well as model-based ones (e.g. EDO) recommend considering a maximum of 0.1 m3/m3 as acceptable tolerance. Having generated also maps of noise, sensitivity for the different products, our results for error variance stay below that threshold generally for most values showing R_TCA values over 0.6 (threshold common in remote sensing studies).

Nonetheless, we consider the thresholds 0.9, 0.6 and 0.4 in R_TCA as the thresholds for minimum, acceptable and optimal values. The three main products involved in the study provide the following percentages of area within the previous thresholds:

EDO: more than 52% of data over R_TCA 0.6, 68% over 0.4, and 12% over 0.9

H121: more than 65% of area over R_TCA 0.6, 67% over 0.4, and 38% over 0.9

CCIp: more than 60% of area over R_TCA 0.6, 70% over 0.4, and 20% over 0.9

We have further specified the % of areas experimenting valid values based on R_TCA and error variance among the different products in several comments along section 4.1.2 (L400-404).

11. The discussion of ISMN network representativity is important but could be expanded to address how spatially limited in-situ data should be used for validating continental-scale products.

Regarding the quality uncertainties of the ISMN, fortunately the database provides flags of for multiple factors of influence beyond the features of SM series themselves (e.g. temperature and rainfall threshold beyond for instance SM series characteristics such as saturation). Data of all stations included in the ISMN database are harmonized and checked for quality control across Europe, and thus the ones included in the study also include these flags. This means that when certain stations whose flags indicated dubious values of SM or of associated variables (such as BIEBRZA network which is known for providing very high SM values) were included based on the criteria of representativity , since in some environmental conditions there is no abundance of stations over all the climate types of Europe. Therefore, it was preferred to deal with uncertainty in a few stations of known factors of uncertainty (e.g. the high values in flood-prone stations like BIEBRZA) than losing the power of valuable yet not optimal SM data from non-key ISMN stations for validating the remote-sensing and model-based products across such a wide range of conditions from boreal to semi-arid conditions.  Therefore, given the few exceptions showing high uncertainty rates under exceptional environmental conditions, flags were waived for the vast majority of stations as it has been already indicated in the manuscript:

*L263-268: "Conversely to the case of the distributed datasets, the point data from the ISMN database has not been severely restricted with flags due to the general consistency of most stations in common environmental conditions and the scarcity of stations with severe indications of uncertainty from the flags. In such cases, the inclusion of the stations was decided based on the existence of alternative stations with similar environmental conditions, and if not available included based on the criteria of consistency between the factors of uncertainty causing the flag and the characteristics of the environment. Multiple areas where snow and icing processes are frequent are barely observed in situ, and consequently, even despite the seasonal uncertainties, RS products provide much more coverage of these areas than the few ISMN stations over these areas. Therefore, including*

*all ISMN networks and data available was the decision adopted to ensure sufficient amount of data for validation over every climatic type and ensure the representativity of a wide range of observed soil moisture conditions."*

Among the factors considered for the selection criteria based on the match between flags and the characteristics of the environment, the land cover was considered. For instance, Fig 2 primarily aimed to illustrate the location of the ISMN networks used for the study while providing secondary information about the dominant land use in such location. The land cover recognition aimed to differentiate if the ISMN locations correspond to areas with canopy (green) or without canopy (yellows) given that vegetation can be the most influential factor over the SM retrievals with remote sensing. In that way, the interpretation of results considered which stations (e.g. primarily some of Scandinavia) were most impacted by dense canopy cover, and accordingly, indicated in the manuscript (L480-481, L514-518).

12. The analysis of trends could benefit from comparison with independent climate variables (precipitation, temperature) to help determine which product trends are more realistic.

Already addressed in the last point of major comments.

13. Page 26, section 6, revise the title for this section regarding the conflict of interest.

The title and text have been revised for clarity.